# Single-electron charge transfer into putative Majorana and trivial modes in individual vortices

Jian-Feng Ge [1], Koen M. Bastiaans[1,2], Damianos Chatzopoulos[1], Doohee Cho [3], Willem O. Tromp[1], Tjerk Benschop [1], Jiasen Niu[1], Genda Gu[4] & Milan P. Allan [1] ✉

Majorana bound states are putative collective excitations in solids that exhibit the self-conjugate property of Majorana fermions—they are their own anti-particles. In iron-based superconductors, zero-energy states in vortices have been reported as potential Majorana bound states, but the evidence remains controversial. Here, we use scanning tunneling noise spectroscopy to study the tunneling process into vortex bound states in the conventional super-conductor $NbSe_2$, and in the putative Majorana platform $FeTe_{0.55}Se_{0.45}$. We find that tunneling into vortex bound states in both cases exhibits charge transfer of a single electron charge. Our data for the zero-energy bound states in $FeTe_{0.55}Se_{0.45}$ exclude the possibility of Yu–Shiba–Rusinov states and are consistent with both Majorana bound states and trivial vortex bound states. Our results open an avenue for investigating the exotic states in vortex cores and for future Majorana devices, although further theoretical investigations involving charge dynamics and superconducting tips are necessary.

When a type-II superconductor is exposed to magnetic fields, vortices emerge as line defects where the order parameter vanishes, quantized magnetic flux penetrates the superconductor, and localized low-energy bound states form in the vortex cores. The nature of the vortex bound state is mysterious in many unconventional superconductors. Recently, much focus is on iron-based superconductors, where topologically nontrivial superconductivity and elusive Majorana bound states have been predicted to exist in vortex cores[1].

So far, the most often reported signature for Majorana bound states is a peak in tunneling differential conductance at zero bias voltage. This signature is readily accessible by experiments, but it is not conclusive proof of the Majorana character of a state[2–5]. Other topologically trivial bound states, including Yu–Shiba–Rusinov (YSR) states, can also show the same zero-bias conductance peak, as demonstrated in proximitized superconducting nanowires[6,7]. Further, Caroli–de Gennes–Matricon (CdGM) states in the vortex cores are

difficult to differentiate from Majorana bound states, because the former could also appear at zero energy[4,8–12].

Zero-bias conductance peaks in full-flux-quantum vortex cores are also the main evidence for Majorana bound states in the iron-based superconductor $FeTe_{0.55}Se_{0.45}$[13], which is the focus of this study. However, controversy remains as the absence of zero-energy bound states has been reported[14,15]. It is still being debated whether the additional observation[16]—a saturating conductance at roughly two-thirds of the expected quantized value $2e^2/h$—is a strong argument for the Majorana character ($h$ is the Planck constant and $e$ is the elementary charge). The issue is that such saturating behavior at an arbitrary conductance near $2e^2/h$ has been observed for YSR states[17] as well; these states are present in $FeTe_{0.55}Se_{0.45}$ and may also appear as a conductance peak at zero bias[18]. Furthermore, it was pointed out that the simple approximations of the Fu–Kane model are not likely applicable to the system of vortices in iron-based superconductors[19],

[1]Leiden Institute of Physics, Leiden University, 2333 CA Leiden, The Netherlands. [2]Department of Quantum Nanoscience, Kavli Institute of Nanoscience, Delft University of Technology, 2628 CJ Delft, The Netherlands. [3]Department of Physics, Yonsei University, Seoul 03722, Republic of Korea. [4]Condensed Matter Physics and Materials Science Department, Brookhaven National Laboratory, Upton, NY 11973, USA. ✉e-mail: allan@physics.leidenuniv.nl

which brings the exact nature of the zero-energy vortex bound states in FeTe$_{0.55}$Se$_{0.45}$ into question.

New local probes are thus desired to investigate the electronic properties of the vortex bound states in iron-based superconductors. It has been widely investigated theoretically how shot noise could act as a tell-tale probe to distinguish between trivial and Majorana bound states in vortex matter and nanowires[20–36], but experiments have not been possible. The principle behind most theoretical proposals is that Majorana bound states induce resonant Andreev reflection: an incident electron from the coupling lead, when tunneling into a Majorana bound state, is reflected as a hole with unity probability[22]. Such a resonant Andreev process is predicted to generate unique Majorana signatures that are absent for trivial fermionic states.

These theoretical studies form the motivation for the shot noise measurements on individual vortex cores that we present here. We measure both the vortex bound states in a conventional superconductor NbSe$_2$ and the putative Majorana bound states in vortices of FeTe$_{0.55}$Se$_{0.45}$ at temperature $T = 2.3$ K. While we argue that our results do not represent a smoking gun experiment for the existence of Majorana bound states, they allow us to exclude YSR states as the origin of the zero-bias conductance peak, and they give an experimental insight into these bound states.

Shot noise is, at its core, a consequence of the discreteness of charge. Because of this, tunneling is a Poissonian process, and the noise spectral density $S$ is proportional to the time-averaged current $I$,

$$S = 2q^*|I|. \tag{1}$$

Shot noise thus allows probing two quantities that are not visible in the time-averaged current: the effective charge $q^*$ of the charge carriers and possible correlations between them in electronic matters[37]. The former has been used to measure fractional charges in mesoscopic quantum hall systems[38], and the latter has been used to measure the vanishing noise at the quantum conductance of break junctions[39].

Despite a large number of theoretical studies on shot noise tunneling into vortex cores, there has been no experiment so far. The challenge is that one needs high enough sensitivity to measure the change in $q^*$ from noise, with nanometer resolution to locate individual vortices. This nanoscale resolution is not feasible in mesoscopic setups where noise measurements have been widely applied.

Recently, we have developed scanning tunneling noise microscopy (STNM), which combines scanning tunneling microscopy (STM) and noise spectroscopy, allowing us to measure the effective tunneling charge with atomic resolution[40]. To do so, we build a cryogenic megahertz amplifier that works in parallel with the usual dc measurements, as illustrated in Fig. 1a. STNM has revealed paired electrons in superconductors[41,42]. STNM also allows to measure shot noise exactly at the core of an individual vortex, which provides a direct and local extraction of the effective charge of the tunneling process into vortex bound states.

Here, we measure two different materials: the iron-based superconductor FeTe$_{0.55}$Se$_{0.45}$, which is conjectured to host putative Majorana bound states, and the conventional superconductor 2H-NbSe$_2$ as a comparison. We use a tip with an apex made out of Pb, which is a type-I, $s$-wave superconductor with a relatively large gap $\Delta_t \sim 1.3$ meV. We choose to use a superconducting tip in this study for two reasons: first, a superconducting tip provides a superior energy resolution without the limitation from thermal broadening as in the case of a normal-metal tip, i.e. ~0.25 meV (See Supplementary Note 1) instead of ~3.5$k_B T = 0.70$ meV, where $k_B$ is the Boltzman constant; second, as a consequence of a convolution with the density of states of the superconducting tip, the tunneling signal into a zero-energy vortex bound state is effectively shifted from the Fermi level to $\pm\Delta_t$ (illustrated

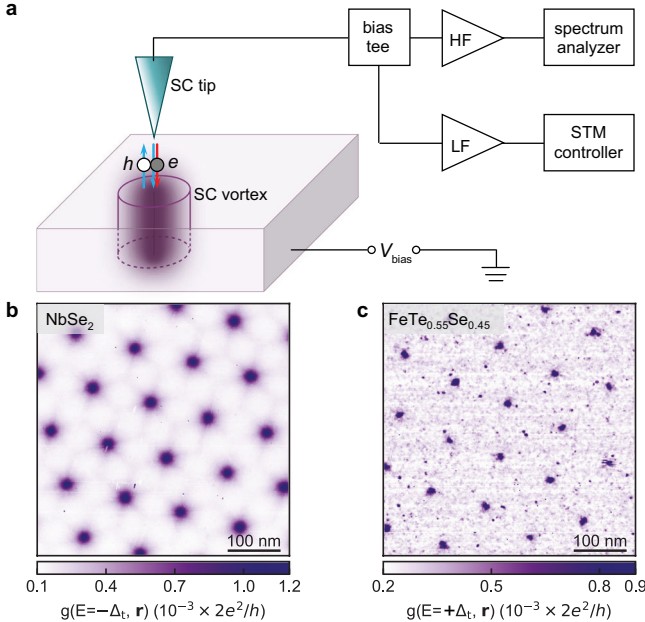

**Fig. 1 | Local tunneling shot noise measurements of vortex bound states.**
**a** Schematic illustration of the scanning tunneling noise microscope setup. A bias voltage ($V_{bias}$) is applied between the superconducting (SC) tip and sample, while the cylinder represents a SC vortex. If the tunneling process is a single-electron (gray) into vortex bound states (red arrow), an effective charge $q^* = 1e$ is transferred from the tip to the vortex. When Andreev reflection takes place (blue arrows), a hole (white) is reflected, and the effective charge doubles $q^* = 2e$. HF and LF stand for the high- and low-frequency amplifier, respectively. STM, scanning tunneling microscope. Full flux quantum ($h/2e$) vortex lattice in NbSe$_2$ (**b**) and FeSe$_{0.55}$Te$_{0.45}$ (**c**) revealed by spatially resolved differential conductance at a magnetic field of 0.1 T. Setup conditions: b, $V_{set} = -5$ mV, $I_{set} = 200$ pA; c, $V_{set} = 10$ mV, $I_{set} = 250$ pA.

in Supplementary Fig. 2)[6]. This shift circumvents the challenge of measuring shot noise at zero bias voltage.

## Results

### Vortex bound states in NbSe$_2$ and FeTe$_{0.55}$Se$_{0.45}$

We first image the subgap electronic structure of the vortices in NbSe$_2$. We introduce vortices by applying an external magnetic field $B = 0.1$ T perpendicular to the sample surface (the critical field of the tip $B_c \sim 0.7$ T, see Supplementary Fig. 1). Because vortices have the strongest enhancement in density of states at the Fermi level of the sample, they are visible as enhanced differential conductance at the energy $|E| = \Delta_t$ when using a superconducting tip (Supplementary Fig. 2a). Figure 1b shows a spatially resolved image of the differential conductance taken with a sample bias $V_{bias} = -\Delta_t/e$, revealing the full-flux-quantum ($h/2e$) vortex lattice, with each vortex in the characteristic sixfold star shape for NbSe$_2$[43,44]. We then take differential conductance maps $g(E, \mathbf{r})$ on a fine spatial grid around an individual vortex as shown in Fig. 2. Away from the vortex core, the spectrum in Fig. 2b shows an energy gap with a size of $2(\Delta_t + \Delta_s)$, where $\Delta_s = 1.0$ meV is the superconducting gap of the sample. On the other hand, the spectrum measured at the core center develops two peaks at $\pm\Delta_t$, which translates to a zero-bias conductance peak for a spectrum taken with a normal-metal tip[6]. This translation is confirmed by a deconvolution procedure[18,45] that extracts the local density of states of the sample (see Supplementary Note 2); as expected, the resulting density of states has a peak at zero energy (Fig. 2e). Deconvolution of spectra along a linecut through the vortex reveals that the zero-bias peak splits away from the core into two dispersing peaks, which eventually merge to the gap edges. These dispersing states are consistent with previous studies[43] of NbSe$_2$ and the expectations of many closely-spaced (on the

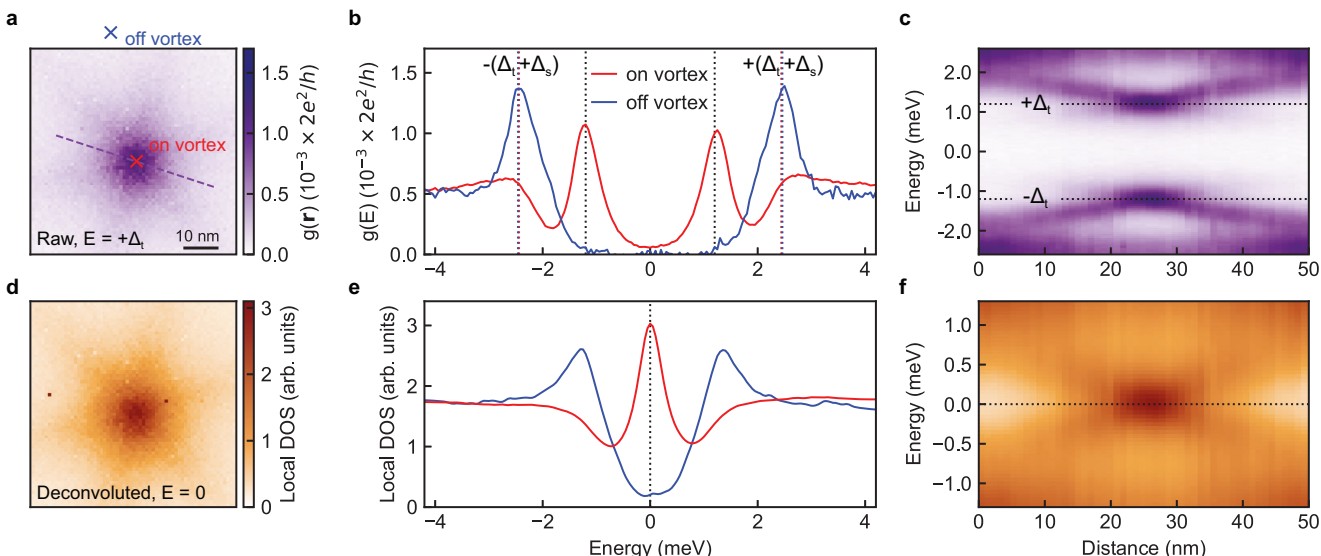

**Fig. 2 | Identifying vortex bound states in NbSe₂ with a superconducting tip.**
**a** Spatially resolved differential conductance around an individual vortex at the energy $E = +\Delta_t$, where $\Delta_t$ and $\Delta_s$ stand for the superconducting gaps of the tip and sample, respectively. **b** High-resolution differential conductance spectra acquired at the two locations marked by the crosses in **a**: the center of the vortex core (red) and off vortex (blue). The off-vortex location is 60 nm away from the core center, in the midpoint between two neighboring vortices of Fig. 1b. The red and blue dashed lines indicate the coherence peaks. **c** Differential conductance spectra along the dashed line (purple) in **a** showing the spatial dispersion of the vortex bound states. The gray dashed lines in **b** and **c** indicate the peaks at $\pm\Delta_t$ where tunneling into the vortex bound states occurs at the core center. **d**–**f** Local density of states (DOS) plots corresponding to **a**–**c**, after deconvolution using the tip DOS. The vortex bound states are indicated by the peak in the local DOS at around zero energy (gray dashed line). **b**, **e** and **c**, **f** share the same horizontal axes. Setup conditions: $V_{set} = 5$ mV, $I_{set} = 200$ pA.

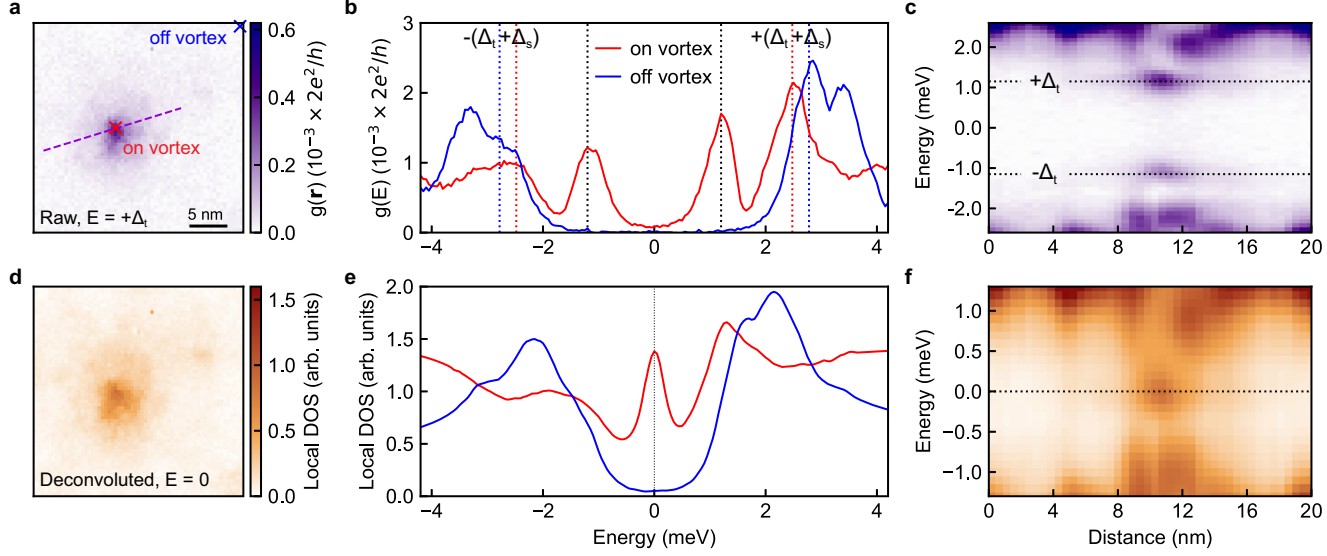

**Fig. 3 | Identifying the putative Majorana bound state in FeTe₀.₅₅Se₀.₄₅.**
**a** Spatially resolved differential conductance around an individual vortex at the energy $E = +\Delta_t$, where $\Delta_t$ and $\Delta_s$ stand for the superconducting gaps of the tip and sample, respectively. **b** High-resolution differential conductance spectra acquired at the two locations marked by the crosses in **a**: the center of the vortex core (red) and off vortex (blue). The red and blue dashed lines indicate the coherence peaks. **c** Differential conductance spectra along the dashed line (purple) in **a** showing the spatial extent of the zero-energy bound state. The gray dashed lines in **b** and **c** indicate the peaks at $\pm\Delta_t$, where tunneling into the putative Majorana bound state occurs. **d**–**f** Local density of states (DOS) plots corresponding to **a**–**c**, after deconvolution using the tip DOS. The putative Majorana bound state is indicated by the peak in the local DOS at zero energy (gray dashed line). **b**, **e** and **c**, **f** share the same horizontal axes. Setup conditions: **a**, **c**, $V_{set} = 10$ mV, $I_{set} = 250$ pA; **b**, $V_{set} = 5$ mV, $I_{set} = 250$ pA.

order of 40 μeV) CdGM bound states from solving the Bogoliubov–de Gennes equations[46], where the peak at a longer distance from the core center corresponds to a CdGM bound state with a larger angular momentum.

In contrast to the dispersing CdGM bound states in vortex cores of NbSe₂, a Majorana bound state is topologically protected such that its energy is locked at the Fermi level[1]. This is exactly what we observe, in

agreement with the literature[13,16], in tunneling differential conductance measurements on vortices of FeTe₀.₅₅Se₀.₄₅ (Fig. 3). The zero-bias conductance peak does not split (Fig. 3c, f) as in the case of NbSe₂ (See Supplementary Note 7); instead, the non-split bound state extends ~8 nm spatially across the vortex core (Supplementary Fig. 2), identical to the states observed and interpreted as Majorana bound states in refs. 13,15,16. In principle, one expects a pair of peaks at $\pm\Delta_t$ with an

equal amplitude in the differential conductance spectrum when tunneling into a Majorana state[47]. We find, that the pairs of peaks $\pm\Delta_t$ we observe at every spectrum on vortex are asymmetric (e.g., the red spectrum in Fig. 3b), which may indicate, the presence of accompanying states such as CdGM or YSR states, assuming a superconducting tip with a particle-hole symmetric density of states. We also note that the hybridization between Majorana bound states in a vortex lattice could also split the conductance peaks owing to the spatial overlap of Majorana wavefunctions. However, since the average distance between vortices in Fig. 1c is about 120 nm, the energy splitting for the putative Majorana bound states is on the order of 1 μeV[48].

Before discussing shot-noise, we comment on what conventional conductance spectroscopy can contribute to distinguishing Majorana, CdGM, or YSR states. CdGM states are expected to be at finite energy instead of zero energy, but the energy difference can be small, and additional effects might shift the energy[8]. CdGM states have been observed in FeTe$_{0.55}$Se$_{0.45}$—surprisingly only in a subset of vortices[13,14]. In these vortices, the lowest energy levels have been reported as small as ~0.1 meV. From Lorentzian fits (Supplementary Fig. 2), our results show that the energy of the zero-bias peak is $0 \pm 50$ μeV, much smaller than the energy of the lowest-lying CdGM bound states reported. Furthermore, a previous high-energy-resolution study[15] has shown that while some vortices show non-zero-bias peaks, associated with CdGM states, the majority (~80% at $B = 1.0$ T) show zero-bias peaks having an energy of $0 \pm 20$ μeV, which is evidence to exclude CdGM states. Based on these studies and the statistics therein, combined with our measured electronic structure, we deduce that the probability that all the vortices measured here only have CdGM states is less than 0.8%. Therefore, the zero-energy state we observe here is in agreement with the putative Majorana bound state previously reported—with the caveat that it has recently been shown that a CdGM state can imitate a zero-energy state[10,12]. We end this discussion by noting that the possibility of trivial YSR states, which can exist at zero bias[18], has been investigated much less.

## Effective charge inside and outside of vortex cores

The key advance of this study is high-sensitivity, atomic-scale noise spectroscopy that allows to extract the effective charge $q^*$ transferred when tunneling into vortex bound states. In the tunneling regime where the transparency is small, we parametrize any changes in noise, including the so-called Fano factor, via the effective charge $q^*$ in Eq. (1) (see Methods for different definitions of the Fano factor). For example, in the simplest case of electron tunneling, the transferred charge of each tunneling event is a single electron charge ($q^* = 1e$), as expected from Poissonian statistics. In contrast, when Andreev reflection takes place such that an incident electron is reflected as a hole, two electron charges ($q^* = 2e$) are effectively transferred per event. Since our measurements are performed at a finite temperature $T$, the current noise in the tunnel junction with resistance $R_J$ consists of shot noise and thermal current noise $4k_BT/R_J$, and takes the form of[37]

$$S = 2q^*(V_{bias}/R_J)\coth(q^*V_{bias}/2k_BT). \quad (2)$$

This equation, which reduces to Eq. (1) at zero temperature, allows us to extract the effective charge $q^*$ as a function of bias voltage. Note that we keep $R_J$ constant during noise spectroscopy by the changing tip-sample distance in a slow feedback loop (See Methods and Supplementary Note 6 for details).

We start by measuring current noise at $B = 0.1$ T at locations far away from the vortex cores. There, one expects the noise to correspond to an effective charge of $q^* = 1e$ at bias energies larger than the gap, $|eV_{bias}| > (\Delta_t + \Delta_s)$. At these energies, the tunneling of Bogoliubov quasiparticles dominates the noise. Around the gap energy $\pm(\Delta_t + \Delta_s)$, one then expects a step in noise from $q^* = 1e$ outside the gap, to $q^* = 2e$ inside. This is because, inside the gap, single-electron processes are

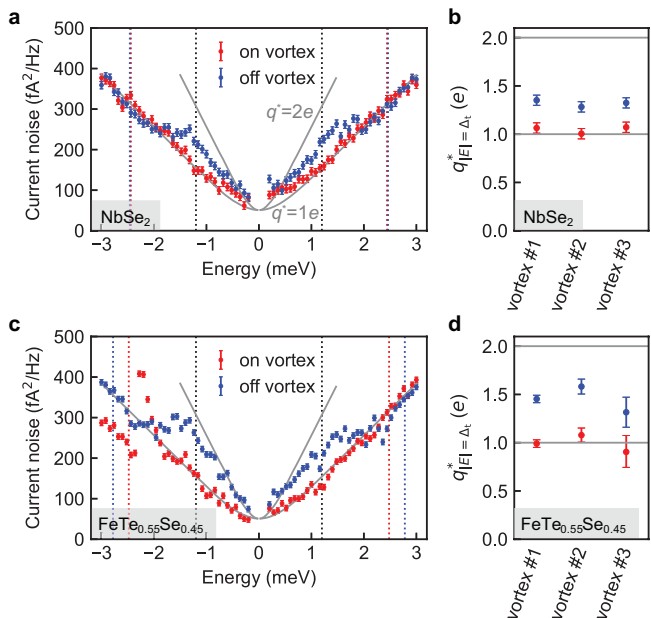

**Fig. 4 | Local noise spectroscopy on and off vortices in NbSe$_2$ and FeTe$_{0.55}$Se$_{0.45}$.**
**a**, **c** Current noise spectra in the tunnel junction (with a resistance $R_J = 2.5$ MOhm) taken on (red) and off (blue) the vortex shown in Fig. 2a for NbSe$_2$ and Fig. 3a for FeTe$_{0.55}$Se$_{0.45}$, respectively. The locations of these spectra are marked by the crosses in Figs. 2a and 3a with the same colors. Gray curves are the expected noise from Eq. 2 with an effective charge $q^*$ of $1e$ and $2e$ at $T = 2.3$ K. The dashed lines in a and c are replicated from Figs. 2b and 3b, respectively, serving as guides for the coherence peaks (red and blue) and the bound states (gray). The error bars are determined by the fluctuation of the current noise in time before each experiment, yielding a standard deviation of 9.25 and 6.77 fA$^2$/Hz for a and c, respectively.
**b**, **d** Effective charge $q^*$ derived by numerically solving Eq. 2 at the energy $E = \pm\Delta_t$ on (red) and off (blue) vortex for three different vortices in NbSe$_2$ and FeTe$_{0.55}$Se$_{0.45}$, respectively. The error bars are determined by the standard deviation of the extracted $q^*$ (Supplementary Figs. 3–5) in the energy ranges ($\Delta_t \pm 0.1$ meV) and -($\Delta_t \pm 0.1$ meV).

not allowed anymore, and only Andreev processes contribute to the noise. As shown in Fig. 4, our measurements are in qualitative agreement with this picture, both in NbSe$_2$ and FeTe$_{0.55}$Se$_{0.45}$. Outside the gap, our data follows the $q^* = 1e$ line, in agreement with Eq. (2). At the gap energy $\pm(\Delta_t + \Delta_s)$, a broadened step is visible in the extracted effective charge spectrum (Supplementary Fig. 5).

Interestingly, $q^*$ does not reach $2e$ inside the gap, but saturates at a value of $1.3e - 1.6e$ (Fig. 4b, d), despite a vanishing conductance within $\pm(\Delta_t + \Delta_s)$ in Figs. 2b and 3b. This is in contrast to the measurement at $B = 0$ T on FeTe$_{0.55}$Se$_{0.45}$, where the extracted $q^*$ reaches $1.97e$ at $\pm\Delta_t$ (see Supplementary Note 4). We hypnotize that the presence of the magnetic field leaves behind a small fraction of delocalized quasiparticles[49], which allows charges of $1e$ to tunnel. Even a very small fraction of quasiparticles will decrease the noise substantially, because for a given tunneling transparency $\tau$, the single-particle processes occur with probability $\tau$, while the Andreev processes occur with probability $\tau^2$ (see Supplementary Note 5 for an estimation of fractions). Future experiments at different magnetic fields, and mapping the exact spatial dependence of current noise around vortex cores are necessary to test this hypothesis relating to the effective charge away from the vortex core. In this study, we focus on the noise spectra in the centers of individual vortex cores.

To investigate the tunneling process into vortex bound states, we then measure the current noise at the vortex cores, first for NbSe$_2$. The experimental data in Fig. 4a show that noise in the core center follows the $1e$-noise behavior until reaching well within $\pm\Delta_t$. We observe a transition from $q^* = 1e$ to $q^* > 1e$ around $\pm1.0$ meV, within which Andreev

reflection at the tip side starts to dominate (Supplementary Fig. 5a). Nevertheless, the $q^*$ remains at $1.05e$ at $E = \pm\Delta_t$, where tunneling into the CdGM bound states occurs (Fig. 4b). As a comparison, we then measured the shot noise of tunneling into the vortex bound states in FeTe$_{0.55}$Se$_{0.45}$. To our surprise, the behaviors of noise and effective charge (Fig. 4c, d) at the vortex cores of FeTe$_{0.55}$Se$_{0.45}$ are very similar to those of NbSe$_2$, i.e., without any Andreev-reflection enhanced noise at $E = \pm\Delta_t$. We extract an effective charge $q^* = 0.99e$ into the zero-energy vortex bound states in FeTe$_{0.55}$Se$_{0.45}$, even closer to a single electron charge than that into CdGM bound states in NbSe$_2$.

## Discussion

We proceed to our discussion of which states are compatible with our results from noise measurements for FeTe$_{0.55}$Se$_{0.45}$. We start with YSR states, which have been observed to cause the zero-bias peak and the saturating conductance, and are present in FeTe$_{0.55}$Se$_{0.45}$[18]. YSR states originate from the resonant coupling between a superconductor and a magnetic impurity. One can tune the coupling by a local gate, such as a voltage-biased STM tip, and the energy levels of the YSR states shift correspondingly to the local field felt by the impurity[18]. Thus, we expect to see spatially dispersing in-gap conductance peaks when moving the tip away from the impurity. The spatial extent of YSR states on FeTe$_{0.55}$Se$_{0.45}$ is about 8 nm (Supplementary Fig. 6), comparable to the size of a vortex. One way is to examine the tunneling process into YSR states, which is expected to be dominated by Andreev reflection in the strong tunneling limit[17]. Naively, the difference compared to the tunneling process into CdGM states can be explained by the different natures of the two states: CdGM states live in the vortex core that extends throughout the superconductor, so that the tunneled electron can leave the superconductor via the one-dimensional vortex core; YSR states localize on the surface of a superconductor around a magnetic impurity, so the tunneled electron cannot go through the superconductor but Andreev-reflected as a hole. Therefore, we expect shot noise with an effective charge of $q^* = 2e$ when tunneling into YSR states[50].

To confirm this picture, we carry out tunneling conductance and noise measurements on the YSR states in FeTe$_{0.55}$Se$_{0.45}$ in the strong tunneling limit (see Supplementary Note 4). The YSR states appear as differential conductance peaks with a ring shape around an impurity site[18]. Our noise measurements (Supplementary Fig. 6) when tunneling into these YSR states show enhanced noise and $q^* \approx 2e$ and are indistinctive from those of tunneling into the bare superconductor. With their stark contrast to the noise and effective charge of $q^* = 1e$ measured for vortex bound states. This leads to the first conclusion of our paper: we can exclude YSR states as the origin of the zero-bias conductance peak.

We then turn to the possibility of Majorana bound states as the origin of the zero-bias peak, as put forward by refs. 13,15,16. Theoretical calculations[20–36] show that shot noise for tunneling into an isolated Majorana bound state vanishes for Majorana-induced Andreev reflection with unity probability, when the bias energy lies within the width of the Majorana bound state. For typical STM measurements where the bias energy is much larger than the intrinsic width of Majorana bound states[16], the tunneling shot noise is Poissonian, i.e., $q^* = 1e$. A second finding is thus that the shot-noise noise we measure is consistent with Majorana bound states.

However, we emphasize that our observation of the identical noise behavior and effective charge for CdGM bound states and the putative Majorana bound states clearly implies consistency with CdGM states as well, at least from a shot-noise point of view. The possible accompanying CdGM states[10] leading to the asymmetry in our differential conductance spectra, would not change the noise behavior but still lead to an effective charge $q^* = 1e$. No theoretical work has focused on tunneling processes into CdGM states. Thus, more theoretical and experimental studies are needed to understand the tunneling process

into CdGM and Majorana bound states in vortices. Still, our work excludes YSR states, and therefore, taken together with high-resolution conductance measurements[15], points towards Majorana modes as likely candidate for the zero-bias peak.

More generally, our work represents a step towards determining the exact nature of a zero-energy state, following theoretical work for Majorana bound states in vortex cores and nanowires. We have measured local shot noise when tunneling into vortex bound states in individual vortices of NbSe$_2$ and FeTe$_{0.55}$Se$_{0.45}$. Using a superconducting tip, we demonstrate the feasibility of measuring shot noise even for states close to the Fermi level, which is usually overwhelmed by thermal noise. First, our data exclude YSR states as the origin of the zero-bias conductance peak at the vortex cores of FeTe$_{0.55}$Se$_{0.45}$. Second, while our data are in agreement with the theoretical prediction for Majorana bound states, we emphasize that we observe an identical shot noise behavior of topologically trivial CdGM bound states in NbSe$_2$.

More theoretical work, especially including a superconducting tip, might allow to gain more information from shot-noise studies. Such a model for the tunneling process from a superconducting tip into a Majorana bound state has already been developed[51], but only in the limit of low temperature. In the future, unambiguous identification of Majorana bound states by shot noise, might be possible in the low-bias limit where temperature and bias energy are both lower than the intrinsic width $\Gamma$ of the zero-bias state, i.e. $eV_{bias} \ll \Gamma$ and $k_B T \ll \Gamma$. This limit can be reached if shot noise measurements are enabled at milli-Kelvin temperatures. There, Majorana-induced resonant Andreev reflection leads to a vanishing shot noise because of unity transmission, which distinguishes itself from $q^* = 1e$ when only CdGM states exist in the vortex core. A further proposal in the low-bias limit[21] suggested a two-tip shot noise measurement setup on two different vortices: each tip tunnels into one localized vortex state, and a positive cross-correlated current noise is expected exclusively for Majorana states. One could further investigate the spin-resolved current-current correlation by a more sophisticated approach combining spin-polarized spectroscopy[52,53] with shot-noise measurements[54,55].

## Methods
### Different definitions of the Fano Factor
While the Fano factor was originally defined as the ratio between the variance and the mean value of a quantity, specific definitions vary in dealing with electrical current and its shot noise. An often applied definition of the Fano factor $F$ is the ratio between the shot noise power $S$ (precisely the Fourier transform of the current-current correlation function) and the Poisson noise $S_P$ due to independent single electrons[37],

$$F = S/S_P = S/2e|I|.$$

In some theory proposals[21,22,27,29] for shot noise of Majorana bound states, one different definition appears where the Fano factor is expressed as

$$F = P/e|I|,$$

where $P$ is the shot noise power (time-averaged current-current correlation function). Another definition of the Fano factor is expressed as the ratio between the differential noise power (the derivative of the time-averaged current-current correlation function with respect to the bias voltage) and the differential conductance[27,28],

$$F = dP/dV/(e \cdot dI/dV).$$

In the above definitions, however, the transmission of a single electron at a time is assumed. As a consequence, the correlation

between them appears as sub- or super-Poissonian shot noise with $F < 1$ or $F > 1$, depending on the details of the transmission probabilities of the conducting channels. In this work, on the other hand, the charge transfer is the quantity of interest, and the STM junction is well in the single-channel, low-transmission regime (our highest tunnel conductance is 0.4 μS, yielding $\tau < 5.2 \times 10^{-3}$). In this regime, we include the possible correlation between charge carriers in the effective charge, $q^* = S/2|I|$, or, more precisely, following Eq. (2).

### Sample preparation and STM measurements

The $FeTe_{0.55}Se_{0.45}$ single crystals with a transition temperature $T_C = 14.5$ K were grown using the Bridgman method. The $2H$-$NbSe_2$ samples ($T_C = 7.2$ K) are purchased from HQ Graphene. The samples with a thickness of ~ 0.5 mm are cleaved in an ultrahigh vacuum at ~30 K and immediately inserted into a customized STM (USM-1500, Unisoku Co., Ltd). All measurements are performed in a cryogenic vacuum at a base temperature of $T = 2.3$ K. We perform scanning tunneling spectroscopy using standard lock-in techniques without the feedback loop enabled. A bias voltage modulation at a frequency of 887 Hz with an amplitude of 100 μV (for maps around vortex) or 50 μV (for high-resolution point spectra) is applied. The resulting differential conductance (d$I$/d$V$) values are normalized by setup conductance $I_{set}/V_{set}$. Prior to all the measurements, a Pt-Ir tip is made superconducting by indenting it into a clean Pb(111) surface. Our superconducting tip exhibits a critical field of about 0.7 T, deducted from differential conductance measurements in different magnetic fields on an atomically flat Au(111) surface (see Supplementary Note 1 for details).

### Noise measurements

We perform noise spectroscopy at a constant junction resistance $R_J$ in a slow feedback loop (see Supplementary Note 6 for details) when varying the bias voltage $V_{bias}$ (and hence tunneling current $I = V_{bias}/R_J$) using our custom-built cryogenic megahertz amplifier developed recently. Because the Josephson tunnel junction with a low $R_J$ may couple to its environment[41,56], which affects the measured noise, we keep our $R_J > 2.5$ MOhm and $V_{bias} > 0.2$ mV, where the Andreev-reflection enhanced conductance at $\pm\Delta_t$ and the environmental coupling effect is negligible. The amplifier consists of an LC tank circuit and a high-electron-mobility transistor that converts the current fluctuations in the junction into voltage fluctuations across a 50 Ohm line, as described in detail elsewhere[40]. To extract the effective charge transferred in the junction we follow a similar procedure as described in refs. 41,42.

The measured total voltage noise is

$$S_V^{meas}(\omega, V) = G^2 |Z_{tot}|^2 S_I,$$

where $G$ is the total gain calibrated by noise spectrum at a high bias (see Supplementary Fig. 9), and $S_I$ is the total current noise

$$S_I = 2q^* I \coth\left(\frac{q^* V}{2k_B T}\right) + \frac{4k_B T R_{res}}{|Z_{tot}|^2} + S_{amp}.$$

The first term is the junction noise from Eq. (2), the second is the thermal noise originating from the resistive part $R_{res}$ of the LC tank circuit, and $S_{amp}$ is the intrinsic current noise of our amplifier.

As the first step of the procedure, we measure the background noise by retracting the tip out of tunneling ($I = 0$ so the first term vanishes), which gives $4k_B T R_{res}/|Z_{res}|^2 + S_{amp}$, where $Z_{tot} = Z_{res}$ because the junction is an open circuit. Then we measure noise in tunneling and subtract it by the background noise (for low $R_J$ we also consider $Z_{tot}$ in the second term as $Z_{res}$ in parallel with $R_J$). Thus, the current noise data plotted in Fig. 4a, c consists only of the noise from

the junction. Finally, we extract the $q^*$ value at each bias by numerically solving Eq. (2).

## Data availability

All data used to generate the figures in the main text and the supplementary information is available on Zenodo at https://doi.org/10.5281/zenodo.7919512.

## Code availability

The code used for this project is available upon request to the authors.

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

## Acknowledgements

We acknowledge C. W. J. Beenakker, C. J. Bolech, T. Hanaguri, T. Machida, D. K. Morr, F. von Oppen, and J. van Ruitenbeek for valuable discussions.

## Author contributions

J.-F.G., K.M.B., D.Cha., D.Cho, W.O.T., T.B., and J.N. performed the experiments and analyzed the data. G.G. grew and characterized the FeTe$_{0.55}$Se$_{0.45}$ samples. All authors contributed to the interpretation of the data and writing of the manuscript. M.P.A. supervised the project.

## Funding

This work was supported by the European Research Council (ERC StG SpinMelt). K.M.B. was supported by the Netherlands Organization for Scientific Research (NWO Veni grant VI.Veni.212.019). D.Cho was supported by the National Research Foundation of Korea (NRF) funded by the Korea government (MSIT) (No. 2020R1C1C1007895 and 2017R1A5A1014862) and the Yonsei University Research Fund of 2019-22-0209. G.G. was supported by the Office of Basic Energy Sciences, Materials Sciences and Engineering Division, US Department of Energy (DOE) under contract number de-sc0012704.

## Competing interests

The authors declare no competing interests.
