## [Peer review file · Nature Communications]

REVIEWER COMMENTS

Reviewer #1 (Remarks to the Author):

This is an excellent experiment complimented by a very well-written manuscript. I believe this manuscript should be accepted immediately, but I do have some small comments that could be addressed in the published draft:

Although the cryostat temperature is indirectly referenced via the thermal broadening listed on page 3, an explicit statement of the temperature would be nice to see early on in the manuscript.

At the beginning of the paper the authors discuss how the inner vortex states show dispersive behavior in the NbSe₂, but not in the FeTeSe. There is an implication that the dispersive behavior points to a CdGM origin for the NbSe₂, and a non-CdGM origin for the FeTeSe. However, at the end of the manuscript the authors do not seem to feel that the dispersivity of the states points to one origin rather than the other, and references 9 and 10 in the manuscript seem to not point to dispersion as a potential indicated for CdGM vs. Majorana modes. The authors should clarify more exactly what is learned from the dispersion measurements, as the side-by-side comparison they perform seems to be more powerful than previous studies. In particular, the authors should indicate how (if at all possible) one material may show dispersion why another may not.

On page 5 it is stated that the high-energy-resolution study [10] can exclude CdGM states as the origin of the zero-bias peak in tunneling spectroscopy data. If this is true, and if this submitted manuscript rules out YSR states as well for the same types of vortices observed in [10], then I wonder why there is not enough evidence yet to confirm the Majorana nature of these bound states. According to the data in [10], the location of a zero-bias bound state does not correlate with the presence of nearby defects and thus the local impurity potential (at least on the surface), so it seems unlikely that all the CdGM states would be shifted to zero bias as proposed by [7]. Of course, a large population of them could be shifted to zero bias, but the large sample size in [10] coupled with this work amounts to very strong evidence for Majoranas in my opinion (modulo any exotic states that have not been considered thus far). I totally understand avoiding making the Majorana claim quite yet, but can the author's clearly state what the biggest reason is for not making such a claim?

On page 6 the authors claim that a nonzero magnetic field leads to a population of delocalized quasiparticles that increase q^* from the expected 1e to the experimentally measured value of 1.3e-1.6e. I would think the presence of these quasiparticles should also lead to an enhanced tunneling

conductance within the gap. Is this observed in that data? If so, it would help justify this theory. Also, it would be nice if the authors provided shot noise data of the NbSe₂ with the field off, to confirm the expected $2q$ dispersion in the absence of magnetic field generated quasiparticles.

On the bottom of page 4 – top of page 5 the authors speculate that the asymmetry of vortex bound states in FeTeSe may be due to ‘accompanying states’. The authors should expand on this discussion and outline which types of states could cause this behavior. Or provide a citation where somebody else has explained it.

Previous measurements by the same group using a Pb tip to measure a Pb surface have shown slightly different behavior than these measurements (10.1103/PhysRevB.100.104506). In particular, states caused by Andreev reflection appear at Δ for reasons unrelated to vortex bound states. Also, a supercurrent peak can be resolved, and peaks due to environmental resonances appear. Can it be assumed that the different tunneling conductances used in those measurements rule out such features appearing in the measurements in this work? If so, the authors should state it clearly.

Finally, I may have missed it, but the authors should state if all measured vortices show the same behavior and, if not, what the statistics are and what other types of behavior are observed (other than YSR states).

Reviewer #2 (Remarks to the Author):

The search for Majorana bound states (MBSs) and their unequivocal demonstration in condensed matter platforms are of great interest now a days. Among the several proposals being intensively explored by the community, one of the most attractive options is based on the Fu-Kane model (three-dimensional strong topological insulator coupled to a conventional s-wave superconductor)

because MBSs are predicted to appear without the need of a Zeeman field or of fine-tuning the chemical potential inside the bulk gap. Recently, it was announced that this model can be realized in a preexisting iron-based superconductor, FeTe_{0.55}Se_{0.45}. Thus, it is of great interest for the community to explore the subgap spectrum of vortices in this material and to try to determine whether zero-bias peaks (ZBPs) some times occurring at the vortex cores are indeed topological MBSs or other trivial zero energy states that may occur in these systems.

Till the moment, the experimental evidence of the existence and properties of zero-energy states is based on the measurement of tunneling conductance ZBPs. In this work, the authors take a step further and measure for the first time local shot noise at the vortex core. This is motivated by the theoretical expectation that shot noise measurements could help to distinguish between trivial and topological states or, at least, to cast further light into the nature of the subgap states. The authors measure local shot noise in individual vortices of both the conventional superconductor NbSe₂ and the putative topological FeTe_{0.55}Se_{0.45}, and compare the results. They find that tunneling into vortex bound states in both cases exhibits a charge transfer of a single electron charge, but there are some interesting differences between both systems. Moreover, they use for their measurements a superconducting tip, which they argue is specially suited to measure shot noise for states close to the Fermi level, which is usually overwhelmed by thermal noise.

I find the subject of the paper of great importance, since as I mentioned above, this is a hot topic for the physics community. Even though it is known that no local measurement can demonstrate unequivocally the presence of (non-local) Majorana modes (due to the existence of quasi-Majoranas or other disorder-induced zero energy modes that are able to mimic the local Majorana phenomenology), it is desirable to subject the purported Majorana states to different tests, of increasing sophistication, to see whether they pass them. These local tests, although not sufficient in general, are necessary conditions to prove the existence of Majoranas, and can very well be used to discard the topological nature of the state under scrutiny. In this sense, the authors are able to contribute with such an extra and valuable test, the shot noise. They find results compatible with the presence of Majoranas in FeTe_{0.55}Se_{0.45}, but they very well admit that other trivial zero energy modes could be responsible of such measurements. I find the paper is very well written and justified, sound, very honest, and surely will attract the interest of the community. Specially, I find that this work will help the Majorana field to advance and other scientist to enroll and contribute to the subject. This is in part because they make a very good case of the importance and developments of the field, with their results included, and also because they expose important open questions, such as for instance the need of theoretical and experimental work focussed on tunneling processes into CdGM states in vortices.

For all these reasons, I confidently recommend this paper for publication in Nature Communications. I have however some questions and comments (some more relevant, others small details) that the authors could consider: ^[1] I think in line 38, where citations [2-4] are given to justify the sentence "This signature is readily accessible by experiments, but it is not conclusive proof of the Majorana character of a state", a further citation should be included because it deals with exactly this problem in a more comprehensive way:

From Andreev to Majorana bound states in hybrid superconductor–semiconductor nanowires

Nature Reviews Physics 2, 575–594 (2020) ^[1] <https://www.nature.com/articles/s42254-020-0228-y>

2) Concerning citations, and due to the closeness in subject, perhaps the authors could also consider citing:

Theory of Caroli-de Gennes-Matricon analogs in full-shell nanowires

<https://arxiv.org/abs/2207.07606>

3) At the level of the discussion of Fig. 1, I think it would be informative to mention why Fig. 1(b) and (c) look so different. I believe this is because NbSe₂ is a perfect non-disordered 2D crystal, whereas in FeTe_{0.55}Se_{0.45} there is some type of surface disorder, which produces all those purple points in between the vortices in the spatially resolved image of the differential conductance. Is this the case? Are perhaps those the YSR states produced by magnetic impurities analyzed further on in the paper?

4) A question about the normalization of the differential conductance in Figs. 2 and 3. Why $g(E)$ is normalized to $g(V_{\text{set}} = 5 \text{ mV})$, instead of to the conductance quantum? Wouldn't this last option be useful to see how far the on-vortex peaks are from conductance quantization?^{[1][1][1][1][1]}
5) A related question: in Figs. 2 and 3 (e) I should understand that the LDOS is in arbitrary units, right?

6) Concerning the differences between Fig. 2(f) and Fig. 3(f): in both cases we can see a zero-energy peak that extends a number of nanometers at the center of the vortex. In particular, the authors observe that for Fig. 3(f) the bound state extends $\sim 8 \text{ nm}$ spatially across the vortex core. Then, in the case of Fig. 2(f), there are other states at higher energies (CdGM states), all the way from zero or close to zero to the gap edge, that are spatially more and more separated from the vortex center and that, as the authors mention, correspond CdGM bound states with larger angular momenta. My question is: why shouldn't I expect to have the same or similar CdGM states in the case of Fig. 3(f)? Do you understand why it seems there is a "gap" between the zero energy state in the case of FeTe_{0.55}Se_{0.45} and the superconducting gap edge? Is there any reason not to have CdGM states in this case?

A further note on this point. Note that the presence or not of other CdGM states shouldn't be in principle a reason to have or not to have Majorana zero modes (or to claim or not to claim Majoranas). At least, I can affirm this in the case of full-shell hybrid nanowires, where my expertise lies. In this case CdGM states may coexist with Majorana zero modes, see "Theory of Caroli-de Gennes-Matricon analogs in full-shell nanowires", <https://arxiv.org/abs/2207.07606>. Moreover, in this case, a zero energy peak (measured at one particular flux value) can be a CdGM state or a Majorana, since CdGM states can reach zero energy when the magnetic field is taken into account inside the vortex. I would expect similar physics for subgap states in superconducting vortices. (Of course, the presence of a gap between the Majorana state and subsequent excited states is desirable to have protected topological states.)

7) Related to the previous point. In lines 138-140 the authors say: "CdGM states are expected to be at finite energy instead of zero energy, but the energy difference can be small, and additional effects might shift the energy[7]."

Whenever I see the expectation that CdGM states should have a finite energy instead of zero energy, I have the same question. Even though it is true that the original paper by Caroli, de Gennes and Matricon in 1964 gives a minimal energy for the lowest Andreev bound state, the famous $\sim \Delta^2/E_F$, soon after (in 1968) Hansen proved that when the magnetic field is considered inside the vortex (something ignored by Caroli et al.), it can take that finite energy to zero. See [7]. The bound excitations of a single vortex in pure type-II superconductor

E. Brun Hansen, Phys. Lett. A 27, 576 (1968). I know that this is perhaps a comment directed more to a theoretical work, but since the authors must be aware of all the latest theory on the type of system they study here, my question is: Shouldn't plain and normal CdGM states be also encountered at zero energy in these vortices? (Perhaps this is studied in their Ref. [7], sorry, I haven't checked). I think the answer is in general yes.

Of course, then we can also have "non-conventional" CdGM states that can be at zero energy or very close, coming from quasi-Majoranas (due to smooth confinement at the surface). This is studied in your Ref. [4]. I think a citation to this very important work in the context of this paper should be also included in your quoted sentence above, together with [7]. And it should also be included in the sentence: "Therefore, the zero-energy state we observe here is in agreement with the putative Majorana bound state previously reported – with the caveat that it has recently been shown that a non-conventional CdGM state might imitate a zero-energy state [7,44]."

8) Another very interesting aspect of this work is the asymmetry the authors observe between the pair of peaks $\pm \Delta t$ in the red spectrum in Fig. 3(b). To be honest, I don't clearly see now whether this would exclude a Majorana interpretation all together, since in principle, one would expect that the resonant Majorana state should couple equally to electrons than to holes from the probe, i.e., positive and negative energies. But perhaps this is not necessary for the peaks at finite energy of Fig. 3(b), as long as the peak at zero energy of the deconvoluted measurement can be considered symmetric in Fig. 3(e) (which I think looks pretty symmetric within the available resolution...).

I know that the subject of conductance asymmetries was studied in this paper by the group of Wimmer:

Conductance asymmetries in mesoscopic superconducting devices due to finite bias

<https://scipost.org/10.21468/SciPostPhys.10.2.037>

Perhaps this cast some light into the subject.

Of course, the question would remain as to why there is not asymmetry in the case of NbSe₂, assuming that you use the same tip and experimental conditions for both systems.

9) In lines 133-136, the authors say: "We first note that the hybridization between Majorana bound states in a vortex lattice could also split the conductance peaks owing to the spatial overlap of Majorana wavefunctions. However, since the average distance between vortices in Fig. 1c is about 120 nm, the energy splitting for the putative Majorana bound states is on the order of 1 μ eV [43]."

When the authors make this comment, I think they refer to Majorana overlap between the, say, upper MBSs of two nearby vortices, i.e., in the (x,y) plane of the (quasi-2D?) superconducting material. But what about Majorana splitting between Majorana partners in the z direction? What is actually the width of the FeTe_{0.55}Se_{0.45} material in the experiment? (Maybe it is said but I missed it, sorry). Do you know what is the theoretical expectation for the MBS decaying length in the z-direction in this system? Is the thickness of the FeTe_{0.55}Se_{0.45} material you measure much larger than this Majorana decaying length? Finally, would it be possible for you to play with the material width, i.e., vortex line length (I mean, in a hypothetical future work) to try and measure Majorana overlap oscillations?

Reviewer #3 (Remarks to the Author):

To Authors

In this manuscript, Ge *et al.* carefully studied and reported on the electron tunneling into the superconductors via the vortex bound states using a local scanning tunneling noise microscopy (STNM). The authors attempt to figure out the nature of the electron transfer not only to the conventional vortex bound states in NbSe₂ but also to the putative Majorana states in FeTe_{0.55}Se_{0.45} which is regarded as the topologically nontrivial vortex bound state. In Majorana research community, it is highly demanding to distinguish the topologically non-trivial states from the other trivial bound states sharing spectral features in the spectroscopic results. In this sense, it is potentially meaningful and desirable to apply STNM to the putative Majorana vortex states accompanied by the comparison with the results on the trivial vortex.

Nevertheless, according to their demonstrations, it seems STNM is not yet suitable to utilize it for the purpose. Significantly, the results obtained even for the conventional vortex bound states are not clear as well, which makes a doubt whether the technique is promising to figure out the relevant physics or not.

Therefore, I cannot recommend this manuscript for the publication in Nature Communications as the current stage based on the concerns as listed below:

Major concerns

(1) The authors claim, "we can exclude YSR states as the origin of the zero-bias conductance peak" as the first conclusion of this manuscript. In general, YSR states appear at the local magnetic impurities in the superconductor. Unless the vortex bound state and YSR state are spatially located with a coincidence, it is easily distinguishable from one to the other based on the presence of the magnetic field as the vortex bound states exist only under the magnetic field. In spite of this fact, the authors have tried to verify the difference in STNM results for two bound states. If there is a specific reason for comparing these two bound states, it should be discussed suitably, for example, in terms of the adatom-induced Majorana vortex as described in Fan *et al.*, Nat. Comm 12, 1348 (2021). By the way, there are no STNM results on a YSR impurity in the supplementary information although the results away from the YSR impurity are presented (Fig. S6e and S6f).

(2) In page 6, from line 169 to 189, they describe the results obtained far away from a vortex (corresponding results are Fig. 4a and 4b, colored in blue). Although they mentioned, "at the gap energy, a step is visible", it is very difficult for me to find any signature of a step in the figures either Fig. 4a, b or Fig. S5a, b at the gap energy. But, there are broad increases at around $\pm(\Delta_T + \Delta_S/2)$.

(3) Although the typical tunneling spectra obtained at a YSR impurity and at off-impurity are clearly distinguishable (Fig. S6d), what would be the benefits to utilize the STNM technique, which does not seem to be sensitive to the localized bound states as demonstrated in Fig. S6b and S6c?

(4) In page 6, line 179: the authors claim q^* reaches $1.97e$ at $\pm\Delta_T$. However, there are significant fractions locating rather close to the line $q^*=1e$ inside a superconducting gap ($\Delta_T \sim \Delta_T + \Delta_S$) for the results away from a YSR impurity (Fig. S6e and f), while the results on Pb(111) from the same group are more convincing for this claim (Ref. 36). What could be the reason for the significant fraction of $q^*=1e$ inside superconducting gap in this material without a magnetic field?

(5) Concerning descriptions for any transitions from $q^*=1$ to $q^*>1$ in this manuscript, it seems that such transitions do not occur at the expected energy positions. On the other hand, the authors tend to stick the q^* values right at the $\pm\Delta_T$ to deduce what they want to emphasize. As the superconducting tip has been used, the substantial contribution of the density of states at the tip can be involved in q^* values at $\pm\Delta_T$ as well. How do they discriminate the contribution of superconducting tip?

(6) The author's hypothesis in line 181 makes the STNM approach doubtful because it is unavoidable to investigate any vortex states without applying a magnetic field, meaning that a charge transfer of $q^*=1e$ contribution always exists inside the superconducting gap. Since the authors provide a significant contribution of the $q^*=1e$ process when two processes coexist as described in supplementary note 5, this is quite serious. It seems this makes even more difficult in the unambiguous distinction between non-trivial states and trivial states.

(7) In this work, the Fano factor is redefined to emphasize the q^* -behavior. On the other hand, in the recent work utilizing the STNM for the YSR states in NbSe₂ (Ref. 46), the coherency of the tunneling processes into the superconductors has been described based on the conventional definition of the Fano factor. I just wonder if the interpretations of the discrepancies from the expected results in this study can be improved or not in terms of the coherency in the tunneling process.

(8) The authors mention that the STM junction is well in the single-channel. How do they confirm this since the valence orbitals contributing to the number of channels in the tunneling into both NbSe₂ and FeTeSe might be different?

Minor comments:

(9) Line 152, Ref. 44 was cited as an example for the existence of mimicking zero-energy states in the non-conventional CdGM state. However, Ref. 44 is the study on the conventional CdGM states in a conventional superconductor (La).

(10) Line 262, As STM experimentalists, do the authors seriously consider the proposal of a two-tip shot noise measurement setup where each tip tunnels into one localized vortex state? Please note that the application of the multi probe STM technique even for the nanostructures on the surface is almost impossible at present if the targeted areas are separated by a distance less than the physical radius of the individual probe. The authors should provide the outlook which can be challenging. Combining the shot-noise experiment with the spin-polarized STM technique (Wang et al., PRL 126, 076802 (2021) and A. Burtzloff et al., PRL 114, 016602 (2015)) would be a more reliable approach, which may be possibly complemented with each other.

Point-by-point reply to the Reviewers' comments

We are grateful that the reviewers point out the novelty and the importance of our approach and results. We thank all reviewers for their helpful comments, and Reviewer #1 and Reviewer #2 for endorsing publication in Nature Communications. We could answer/rebut all of Reviewer #3's comments, and were able to provide all of the data that they requested. We thank all reviewers for helping us steer the discussion in the paper to a better version. Below we reply to all comments in detail.

We marked all new changes for this revision in the manuscript in blue. Below is a summary of the changes:

- Main text: we added two references (Refs. 5 and 11) suggested by Reviewer #2.
- Main text, Page 2, first paragraph: we changed the last sentence that CdGM states can appear at zero energy as suggested by Reviewer #2.
- Main text, Page 2, last paragraph: we added our measurement temperature suggested by Reviewer #1.
- Main text, Page 4, last paragraph: we changed a sentence to include examples of 'accompanying states', as suggested by Reviewer #1.
- Main text, Page 6, 2nd paragraph: we changed the last sentence to be more accurate about the 'step', as suggested by Reviewer #3
- Main text, Page 7, 2nd paragraph: we added a sentence about the reason for comparing YSR and CdGM states, as suggested by Reviewer #3.
- Main text, Page 8, last paragraph: we added a sentence about future experiments using spin-polarized shot noise, as suggested by Reviewer #3.
- Main text (Methods), Page 9, last sentence: we added information of sample thickness, as suggested by Reviewer #2.
- Main text (Methods), Page 10, 2nd paragraph: we changed a sentence about negligible Andreev peaks, as suggested by Reviewer #1.
- All Figures: we changed the normalization of differential conductance to quantized conductance $2e^2/h$, as suggested by Reviewer #2.
- Supplemental Materials: we added Note 7 to explain the difference in dispersion of CdGM states in $\text{FeTe}_{0.55}\text{Se}_{0.45}$ and NbSe_2 , as suggested by Reviewer #1.
- Supplemental Materials: we added Fig. S10 to show $q^*=2e$ on NbSe_2 at zero field, as requested by Reviewer #1.

Additionally, we performed small changes marked in blue in the main text, such as rewording sentences and correcting grammar errors.

With kind regards,

Milan Allan, on behalf of the authors

Point-by-point reply to Reviewers.

Reviewer #1

This is an excellent experiment complimented by a very well-written manuscript. I believe this manuscript should be accepted immediately, but I do have some small comments that could be addressed in the published draft:

We thank the reviewer for recommending immediate publication of our work, and for the helpful comments.

Although the cryostat temperature is indirectly referenced via the thermal broadening listed on page 3, an explicit statement of the temperature would be nice to see early on in the manuscript.

We have added this information ($T = 2.3$ K) at the end of page 2.

At the beginning of the paper the authors discuss how the inner vortex states show dispersive behavior in the NbSe₂, but not in the FeTe_{0.55}Se_{0.45}. There is an implication that the dispersive behavior points to a CdGM origin for the NbSe₂, and a non-CdGM origin for the FeTe_{0.55}Se_{0.45}. However, at the end of the manuscript the authors do not seem to feel that the dispersivity of the states points to one origin rather than the other, and references 9 and 10 in the manuscript seem to not point to dispersion as a potential indicated for CdGM vs. Majorana modes. The authors should clarify more exactly what is learned from the dispersion measurements, as the side-by-side comparison they perform seems to be more powerful than previous studies. In particular, the authors should indicate how (if at all possible) one material may show dispersion why another may not.

For a conventional s-wave BCS superconductor, CdGM states have been extensively studied theoretically (e.g. Ref. 46): CdGM states have an increasing angular momentum when moving a distance r away from the vortex core. As a consequence, the majority of the CdGM states that contribute to the differential conductance have an energy E_p approximately proportional to $k_F \cdot r$, where k_F is the Fermi wavevector. In addition, at E_p the differential conductance maximum decays exponentially in r on a length scale of coherence length ξ . Therefore, the dispersion profile of CdGM states depends crucially on two material parameters k_F and ξ (see Table. R1 for their values of NbSe₂ and FeTe_{0.55}Se_{0.45}). For NbSe₂, both parameters are larger, and the dispersion is measurable by STM. However, for FeTe_{0.55}Se_{0.45}, k_F is one order of magnitude smaller so E_p changes much more slowly with r ; meanwhile ξ is also smaller, resulting in a vanishing amplitude before E_p changes significantly. A similar discussion can be found in Ref. 14. Therefore, observation of dispersing states indicates a CdGM origin, but non-dispersing states cannot exclude a CdGM origin.

For this reason, we did not claim a non-CdGM origin for FeTe_{0.55}Se_{0.45} purely based on the non-dispersing feature. As we stated on Page 5, based on the statistics from the previous high-resolution STM results (Ref. 15), there is still a low, but finite possibility that they could be CdGM states. If they were CdGM states, their energy is 0 ± 50 ueV, much lower than the *observed* lowest-lying CdGM states at 0.1 meV. On the other hand, we cannot exclude a CdGM origin as theoretically proposed by Refs. [4, 8-12].

We have added a paragraph in the supplementary (Supplementary Note 7) to clarify this important issue.

Table R1. Fermi wavevector and coherence length of NbSe₂ and FeTe_{1-x}Se_x.

Material	NbSe ₂	FeTe _{1-x} Se _x
Fermi wavevector k_f	0.5~1 Å ⁻¹ [PRB.85.224532]	0.07~0.12 Å ⁻¹ [Ref. 14]
Coherence length ξ	12 nm [PRB.21.2717]	3 nm [PRB.81.094518]

On page 5 it is stated that the high-energy-resolution study [10] can exclude CdGM states as the origin of the zero-bias peak in tunneling spectroscopy data. If this is true, and if this submitted manuscript rules out YSR states as well for the same types of vortices observed in [10], then I wonder why there is not enough evidence yet to confirm the Majorana nature of these bound states. According to the data in [10], the location of a zero-bias bound state does not correlate with the presence of nearby defects and thus the local impurity potential (at least on the surface), so it seems unlikely that all the CdGM states would be shifted to zero bias as proposed by [7]. Of course, a large population of them could be shifted to zero bias, but the large sample size in [10] coupled with this work amounts to very strong evidence for Majoranas in my opinion (modulo any exotic states that have not been considered thus far). I totally understand avoiding making the Majorana claim quite yet, but can the author's clearly state what the biggest reason is for not making such a claim?

We agree with the reviewer that based on Ref. 10 (Now Ref. 15 in the revised manuscript) and our exclusion of YSR states, Majorana bound states are the prime candidate for the origin of the zero-bias peak from the current understanding. We do not want to make this claim yet, because (1) vortex bound states in unconventional superconductors are more complicated than the dichotomy between Majorana and trivial CdGM states; there might be other exotic states in these vortices unknown to present theory. (2) local measurements can only select candidates for Majorana bound states by their appearance (e.g. a zero-bias peak). We believe that one can make a claim of Majorana states based on the nonlocal and topological properties, such as topological quantum phase transition in nonlocal tunneling (See <https://doi.org/10.1038/s41567-022-01900-9> for a very recent perspective article in *Nature Physics* by Das Sarma), and ultimately non-Abelian braiding statistics. See also the last paragraph on Page 8 in our text.

On page 6 the authors claim that a nonzero magnetic field leads to a population of delocalized quasiparticles that increase q^* from the expected $1e$ to the experimentally measured value of $1.3e$ - $1.6e$. I would think the presence of these quasiparticles should also lead to an enhanced tunneling conductance within the gap. Is this observed in that data? If so, it would help justify this theory. Also, it would be nice if the authors provided shot noise data of the NbSe₂ with the field off, to confirm the expected $2q$ dispersion in the absence of magnetic field generated quasiparticles.

We agree that the presence of delocalized quasiparticles should, in principle, increase the tunneling conductance. However, based on the resulting $q^* = 1.3e \sim 1.6e$, we estimate the increment is in the range of $0.1 \sim 1\%$ (see Fig. S7), which is beyond our resolution for tunneling conductance inside the gap. We have added zero-field shot noise data measured on NbSe₂ in Supplementary Figure 10 and as shown below. It clearly shows a $q^*=2e$ inside the gap in the absence of the magnetic field. We did not claim but just

speculate that the magnetic field generates delocalized quasiparticles (see e.g. Ref. 54 and Science 374, 1381), which can lower q^* below $2e$. This issue is under our more careful investigation at the moment.

Figure R1. Differential conductance and noise spectra measured on NbSe₂ at zero field. **a** Differential conductance spectrum and **b** corresponding noise spectrum measured at $B = 0$ T and a random location on a NbSe₂ sample. Noise data increase from $q^*=1e$ curve starting at $\pm(\Delta_t + \Delta_s)$, towards to $q^*=2e$ curve inside the gap. Setup conditions: **a**, $V_{\text{set}} = -5$ mV, $I_{\text{set}} = 200$ pA; **b**, $R_I = 2.5$ MOhm.

On the bottom of page 4 – top of page 5 the authors speculate that the asymmetry of vortex bound states in FeTeSe may be due to ‘accompanying states’. The authors should expand on this discussion and outline which types of states could cause this behavior. Or provide a citation where somebody else has explained it.

We apologize for not being clear in the discussion on the asymmetry. Electron-hole symmetry in the conductance spectra obtained by a superconducting tip has been argued as a signature for Majorana bound states in atom chains (Ref. 47), while the asymmetry may imply trivial state contribution, e.g. from CdGM states or YSR states as discussed in the manuscript, around Fermi level of the sample.

We have revised this sentence in the main text to make it clearer.

Previous measurements by the same group using a Pb tip to measure a Pb surface have shown slightly different behavior than these measurements (10.1103/PhysRevB.100.104506). In particular, states caused by Andreev reflection appear at $\pm\Delta_T$ for reasons unrelated to vortex bound states. Also, a supercurrent peak can be resolved, and peaks due to environmental resonances appear. Can it be assumed that the different tunneling conductances used in those measurements rule out such features appearing in the measurements in this work? If so, the authors should state it clearly.

Yes, the Andreev-reflection enhanced conductance at $\pm\Delta_t$ is only visible at very low junction resistance below 1 MOhm, and their amplitudes are much smaller than the coherence peaks at $\pm(\Delta_t+\Delta_s)$. For the experiments presented in this work, the junction resistance is always above 2.5 MOhm so we do not expect to see the Andreev-reflection enhanced conductance. Also note that the peaks at $\pm\Delta_t$ in the vortex cores (red curves in Figs. 2b&3b) is on the same order or even higher than normal-state conductance, which is not the case for Andreev-reflection enhanced conductance for a Pb tip-Pb sample setup at this junction resistance. Based on those facts, we can exclude Andreev process as the origin of the enhanced conductance at $\pm\Delta_t$.

This discussion is already in the Methods section and we have revised it accordingly to be clearer.

Finally, I may have missed it, but the authors should state if all measured vortices show the same behavior and, if not, what the statistics are and what other types of behavior are observed (other than YSR states).

We showed the measured effective charge for three vortices of NbSe₂ in Fig. 4b and three vortices of FeTe_{0.55}Se_{0.45} in Fig. 4d. The detailed tunneling conductance and shot noise data are presented in Figs. S3 & S4 for NbSe₂ and FeTe_{0.55}Se_{0.45}. Within each material, all three vortices show the same behavior.

Reviewer #2

The search for Majorana bound states (MBSs) and their unequivocal demonstration in condensed matter platforms are of great interest now a days. Among the several proposals being intensively explored by the community, one of the most attractive options is based on the Fu-Kane model (three-dimensional strong topological insulator coupled to a conventional s-wave superconductor)

because MBSs are predicted to appear without the need of a Zeeman field or of fine-tuning the chemical potential inside the bulk gap. Recently, it was announced that this model can be realized in a preexisting iron-based superconductor, FeTe_{0.55}Se_{0.45}. Thus, it is of great interest for the community to explore the subgap spectrum of vortices in this material and to try to determine whether zero-bias peaks (ZBPs) some times occurring at the vortex cores are indeed topological MBSs or other trivial zero energy states that may occur in these systems.

Till the moment, the experimental evidence of the existence and properties of zero-energy states is based on the measurement of tunneling conductance ZBPs. In this work, the authors take a step further and measure for the first time local shot noise at the vortex core. This is motivated by the theoretical expectation that shot noise measurements could help to distinguish between trivial and topological states or, at least, to cast further light into the nature of the subgap states. The authors measure local shot noise

in individual vortices of both the conventional superconductor NbSe₂ and the putative topological FeTe_{0.55}Se_{0.45}, and compare the results. They find that tunneling into vortex bound states in both cases exhibits a charge transfer of a single electron charge, but there are some interesting differences between both systems. Moreover, they use for their measurements a superconducting tip, which they argue is specially suited to measure shot noise for states close to the Fermi level, which is usually overwhelmed by thermal noise.

I find the subject of the paper of great importance, since as I mentioned above, this is a hot topic for the physics community. Even though it is known that no local measurement can demonstrate unequivocally the presence of (non-local) Majorana modes (due to the existence of quasi-Majoranas or other disorder-induced zero energy modes that are able to mimic the local Majorana phenomenology), it is desirable to subject the purported Majorana states to different tests, of increasing sophistication, to see whether they pass them. These local tests, although not sufficient in general, are necessary conditions to prove the existence of Majoranas, and can very well be used to discard the topological nature of the state under scrutiny. In this sense, the authors are able to contribute with such an extra and valuable test, the shot noise. They find results compatible with the presence of Majoranas in FeTe_{0.55}Se_{0.45}, but they very well admit that other trivial zero energy modes could be responsible of such measurements. I find the paper is very well written and justified, sound, very honest, and surely will attract the interest of the community. Specially, I find that this work will help the Majorana field to advance and other scientist to enroll and contribute to the subject. This is in part because they make a very good case of the importance and developments of the field, with their results included, and also because they expose important open questions, such as for instance the need of theoretical and experimental work focussed on tunneling processes into CdGM states in vortices.

For all these reasons, I confidently recommend this paper for publication in Nature Communications.

We thank the Reviewer for recommending the publication of our work, for the nice summary, the positive assessment, and for the helpful comments below.

I have however some questions and comments (some more relevant, others small details) that the authors could consider:

1) I think in line 38, where citations [2-4] are given to justify the sentence "This signature is readily accessible by experiments, but it is not conclusive proof of the Majorana character of a state", a further citation should be included because it deals with exactly this problem in a more comprehensive way: From Andreev to Majorana bound states in hybrid superconductor–semiconductor nanowires Nature Reviews Physics 2, 575–594 (2020) <https://www.nature.com/articles/s42254-020-0228-y>

2) Concerning citations, and due to the closeness in subject, perhaps the authors could also consider citing: Theory of Caroli-de Gennes-Matricon analogs in full-shell nanowires <https://arxiv.org/abs/2207.07606>

We thank the reviewer for the above references, and we have added them as Ref. 5 and Ref. 11 in the revised manuscript.

3) At the level of the discussion of Fig. 1, I think it would be informative to mention why Fig. 1(b) and (c) look so different. I believe this is because NbSe₂ is a perfect non-disordered 2D crystal, whereas in FeTe_{0.55}Se_{0.45} there is some type of surface disorder, which produces all those purple points in between the vortices in the spatially resolved image of the differential conductance. Is this the case? Are perhaps those the YSR states produced by magnetic impurities analyzed further on in the paper?

Yes, NbSe₂ has much less disorder than FeTe_{0.55}Se_{0.45}, so the vortex lattice appears perfect in NbSe₂, while the FeTe_{0.55}Se_{0.45} is known to be inhomogeneous (including chemical disorder and superfluid density, see our previous work Ref. 52). The scattered purple points in Fig. 1(c) indicate states near $E = \pm \Delta_v$, and they can be impurity states, including YSR states, but they can also be of other, unknown origin.

4) A question about the normalization of the differential conductance in Figs. 2 and 3. Why $g(E)$ is normalized to $g(V_{\text{set}} = 5 \text{ mV})$, instead of to the conductance quantum? Wouldn't this last option be useful to see how far the on-vortex peaks are from conductance quantization?

We have adopted the reviewer's suggestion to use the common conductance quantum $2e^2/h$ as the normalization. We note that, however, it is not clear what conductance quantization one expects for a superconducting tip tunneling into a vortex (or Majorana) bound state. Theoretical calculation (Ref. 51) shows that using a superconducting lead the Majorana quantization yields $(4-\pi) * 2e^2/h$ instead of $2e^2/h$, but this has yet to be confirmed experimentally.

5) A related question: in Figs. 2 and 3 (e) I should understand that the LDOS is in arbitrary units, right?

Yes, the LDOS of the sample is in arbitrary units after the deconvolution procedure (see Supplementary Note 2 for details). We have added 'arb. unit' in Figs. 2&3 for LDOS.

6) Concerning the differences between Fig. 2(f) and Fig. 3(f): in both cases we can see a zero-energy peak that extends a number of nanometers at the center of the vortex. In particular, the authors observe that for Fig. 3(f) the bound state extends $\sim 8 \text{ nm}$ spatially across the vortex core. Then, in the case of Fig. 2(f), there are other states at higher energies (CdGM states), all the way from zero or close to zero to the gap edge, that are spatially more and more separated from the vortex center and that, as the authors mention, correspond CdGM bound states with larger angular momenta. My question is: why shouldn't I expect to have the same or similar CdGM states in the case of Fig. 3(f)? Do you understand why it seems there is a "gap" between the zero energy state in the case of FeTe_{0.55}Se_{0.45} and the superconducting gap edge? Is there any reason not to have CdGM states in this case?

Indeed, one does expect CdGM states in FeTe_{0.55}Se_{0.45}. However, identifying them is rather complicated, as briefly discussed in Ref. 14. First, as the energy of the lowest-lying CdGM states scales with Δ_s^2/E_F , for FeTe_{0.55}Se_{0.45} this yields $0.22 \sim 0.74 \text{ meV}$ for the lowest-lying CdGM states (Refs. 13-15). A local change Δ_s or E_F could even shift it to even higher energy. Considering Δ_s on the order of 1.5 meV , only the first 2-5 CdGM levels are expected, compared to much more of them in NbSe₂ ($\Delta_s^2/E_F \sim 40 \text{ ueV}$ and $\Delta_s \sim 1.3 \text{ meV}$). In Ref. 14, the first 3 CdGM levels are reported based on their equal spacing, but only in a fraction ($\sim 20\%$) of vortices in FeTe_{0.55}Se_{0.45}. In the rest vortices, only a single peak is observed in the gapped spectrum, which is not sufficient to identify them as CdGM states, as YSR states induced by local defects can also produce the same spectrum. Second, similar to the 2nd point of Reviewer #1, the dispersion profile of

CdGM states depends crucially on two material parameters: Fermi wavelength k_F and coherence length ξ (see Table. R1 for their values of NbSe₂ and FeTe_{0.55}Se_{0.45}). For NbSe₂, both parameters are bigger, and the dispersion is measurable by STM. However, for FeTe_{0.55}Se_{0.45}, k_F is one order of magnitude smaller so the CdGM states disperse slowly, as observed in Refs. 14-16; meanwhile ξ is also smaller, i.e. the maximal amplitude vanishes before the energy changes significantly. In summary, the rather small number of CdGM states and their slow dispersion expected in FeTe_{0.55}Se_{0.45} would result in CdGM states having in principle very different appearance from those in NbSe₂. We have included the latter part of the discussion in Supplementary Note 7.

We do not know the reason why there is a “gap” between the zero-energy state in the case of FeTe_{0.55}Se_{0.45} and the superconducting gap edge, but in Refs. 13 and 15, a clear decrease in differential conductance at the superconducting gap edges was also observed where the zero-bias states appear.

A further note on this point. Note that the presence or not of other CdGM states shouldn't be in principle a reason to have or not to have Majorana zero modes (or to claim or not to claim Majoranas). At least, I can affirm this in the case of full-shell hybrid nanowires, where my expertise lies. In this case CdGM states may coexist with Majorana zero modes, see "Theory of Caroli-de Gennes-Matricon analogs in full-shell nanowires", <https://arxiv.org/abs/2207.07606>. Moreover, in this case, a zero energy peak (measured at one particular flux value) can be a CdGM state or a Majorana, since CdGM states can reach zero energy when the magnetic field is taken into account inside the vortex. I would expect similar physics for subgap states in superconducting vortices. (Of course, the presence of a gap between the Majorana state and subsequent excited states is desirable to have protected topological states.)

We agree with the reviewer that CdGM and Majorana states are not mutually exclusive. Therefore, we tried to be careful in making a claim regarding the issue. We derive, based on the statistics of the previous high-resolution STM study (Ref. 15), that the possibility for non-zero-bias peaks, in all three vortices we measured in FeTe_{0.55}Se_{0.45}, is lower than 0.8%. Ref. 15 uses the energy argument, as mentioned in the above point that the lowest *observed* CdGM states lie above 0.1 meV, to exclude CdGM states. However, as the reviewer pointed out, the exclusion cannot be simply made once when the magnetic field inside the vortex is taken into account. See our reply for the following point.

7) Related to the previous point. In lines 138-140 the authors say: "CdGM states are expected to be at finite energy instead of zero energy, but the energy difference can be small, and additional effects might shift the energy[7]."

Whenever I see the expectation that CdGM states should have a finite energy instead of zero energy, I have the same question. Even though it is true that the original paper by Caroli, de Gennes and Matricon in 1964 gives a minimal energy for the lowest Andreev bound state, the famous $\sim \Delta^2/E_F$, soon after (in 1968) Hansen proved that when the magnetic field is considered inside the vortex (something ignored by Caroli et al.), it can take that finite energy to zero. See: The bound excitations of a single vortex in pure type-II superconductor E. Brun Hansen, Phys. Lett. A 27, 576 (1968). I know that this is perhaps a comment directed more to a theoretical work, but since the authors must be aware of all the latest theory on the type of system they study here, my question is: Shouldn't plain and normal CdGM states be also encountered at zero energy in these vortices? (Perhaps this is studied in their Ref. [7], sorry, I haven't checked). I think the answer is in general yes.

We thank the reviewer for providing the reference. We agree that, in general, when the Zeeman effect is taken into consideration, in certain situations one spin specie of the lowest CdGM states can shift to the Fermi level. We have changed our sentence in the main text and included this reference as Ref. 8, together with a more recent one (Ref. 9). Nevertheless, it is worth noting that the Zeeman correction term is magnetic field dependent, so this coincidence of zero-energy CdGM states only occurs at a certain field H_z (and thus a certain external field B), and appears in every single vortex core in the absence of disorder.

Of course, then we can also have "non-conventional" CdGM states that can be at zero energy or very close, coming from quasi-Majoranas (due to smooth confinement at the surface). This is studied in your Ref. [4]. I think a citation to this very important work in the context of this paper should be also included in your quoted sentence above, together with [7]. And it should also be included in the sentence: "Therefore, the zero-energy state we observe here is in agreement with the putative Majorana bound state previously reported – with the caveat that it has recently been shown that a non-conventional CdGM state might imitate a zero-energy state[7,44]."

Ref. 4 studied Andreev bound states which have a different origin from the CdGM states. CdGM states are created by the phase winding around a vortex, while Andreev bound states, well known in nanowires, are localized quasiparticles bound to disorder. Andreev bound states can appear in $\text{FeTe}_{0.55}\text{Se}_{0.45}$, and even near a vortex core, but Ref. 15 showed that the appearance of a zero-bias peak does not correlate to chemical disorder on the surface of $\text{FeTe}_{0.55}\text{Se}_{0.45}$. Therefore, in the discussion about CdGM states, we do not include Ref. 4 related to Andreev bound states.

8) Another very interesting aspect of this work is the asymmetry the authors observe between the pair of peaks $\pm\Delta t$ in the red spectrum in Fig. 3(b). To be honest, I don't clearly see now whether this would exclude a Majorana interpretation all together, since in principle, one would expect that the resonant Majorana state should couple equally to electrons than to holes from the probe, i.e., positive and negative energies. But perhaps this is not necessary for the peaks at finite energy of Fig. 3(b), as long as the peak at zero energy of the deconvoluted measurement can be considered symmetric in Fig. 3(e) (which I think looks pretty symmetric within the available resolution...).

I know that the subject of conductance asymmetries was studied in this paper by the group of Wimmer: Conductance asymmetries in mesoscopic superconducting devices due to finite bias <https://scipost.org/10.21468/SciPostPhys.10.2.037> Perhaps this cast some light into the subject.

Of course, the question would remain as to why there is not asymmetry in the case of NbSe_2 , assuming that you use the same tip and experimental conditions for both systems.

Electron-hole symmetry in the conductance spectra obtained by a *superconducting* tip has been argued as a signature for Majorana bound states in atomic chains (Ref. 47), while an asymmetry may imply trivial state contribution around Fermi level of the sample, but can also be a consequence of tip effects. Since the tip for measuring NbSe_2 sample is not exactly identical to the tip for $\text{FeTe}_{0.55}\text{Se}_{0.45}$, we choose not to make strong statements about which aspects influence the asymmetry.

9) In lines 133-136, the authors say: "We first note that the hybridization between Majorana bound states in a vortex lattice could also split the conductance peaks owing to the spatial overlap of Majorana

wavefunctions. However, since the average distance between vortices in Fig. 1c is about 120 nm, the energy splitting for the putative Majorana bound states is on the order of $1 \mu\text{eV}$ [43]."

When the authors make this comment, I think they refer to Majorana overlap between the, say, upper MBSs of two nearby vortices, i.e., in the (x,y) plane of the (quasi-2D?) superconducting material. But what about Majorana splitting between Majorana partners in the z direction? What is actually the width of the FeTe_{0.55}Se_{0.45} material in the experiment? (Maybe it is said but I missed it, sorry). Do you know what is the theoretical expectation for the MBS decaying length in the z-direction in this system? Is the thickness of the FeTe_{0.55}Se_{0.45} material you measure much larger than this Majorana decaying length? Finally, would it be possible for you to play with the material width, i.e., vortex line length (I mean, in a hypothetical future work) to try and measure Majorana overlap oscillations?

Yes, we meant the hybridization between Majorana bound states in two neighboring vortices on the surface. The localization length of Majorana bound states in iron-based superconductors, according to Ref. 1, extends on the order of 100 nm in z direction. Since our sample thickness is about 0.5 mm, we can safely neglect the overlap of Majorana states in z direction. We have added this information (thickness) in the Methods section.

For future work on Majorana overlap oscillations: it is rather challenging for STM to access the DOS profile along a vortex line, but there are proposals to measure in-plane oscillation via Josephson STM [e.g. PhysRevB.98.134502], which is technically much easier.

Reviewer #3

In this manuscript, Ge et al. carefully studied and reported on the electron tunneling into the superconductors via the vortex bound states using a local scanning tunneling noise microscopy (STNM). The authors attempt to figure out the nature of the electron transfer not only to the conventional vortex bound states in NbSe₂ but also to the putative Majorana states in FeTe_{0.55}Se_{0.45} which is regarded as the topologically nontrivial vortex bound state. In Majorana research community, it is highly demanding to distinguish the topologically non-trivial states from the other trivial bound states sharing spectral features in the spectroscopic results. In this sense, it is potentially meaningful and desirable to apply STNM to the putative Majorana vortex states accompanied by the comparison with the results on the trivial vortex.

Nevertheless, according to their demonstrations, it seems STNM is not yet suitable to utilize it for the purpose. Significantly, the results obtained even for the conventional vortex bound states are not clear as well, which makes a doubt whether the technique is promising to figure out the relevant physics or not.

Therefore, I cannot recommend this manuscript for the publication in Nature Communications as the current stage based on the concerns as listed below:

We thank the reviewer for appreciating the novelty and technical qualities of our experimental work. We do not agree with the reviewer that STNM will not be suitable for distinguishing between Majorana and trivial states. STNM is a new technique and still undergoes development, especially in terms of resolution

and compatibility with extreme conditions. We have already pointed out in the last paragraph that, once in the strong tunneling, low-bias regime at millikelvin temperature, STNM results are predicted to show a clearer distinction between Majorana and trivial states (see e.g. Ref. 34). In addition, nonlocal tunneling experiments, via a two-tip STNM (Ref. 21), can provide much stronger evidence for Majorana based on their nonlocal properties.

Below we address the reviewer's concerns, and we hope that after our reply and revision, the reviewer is convinced that our work is suitable for publication in Nature Communications.

Major concerns

(1) The authors claim, "we can exclude YSR states as the origin of the zero-bias conductance peak" as the first conclusion of this manuscript. In general, YSR states appear at the local magnetic impurities in the superconductor. Unless the vortex bound state and YSR state are spatially located with a coincidence, it is easily distinguishable from one to the other based on the presence of the magnetic field as the vortex bound states exist only under the magnetic field. In spite of this fact, the authors have tried to verify the difference in STNM results for two bound states. If there is a specific reason for comparing these two bound states, it should be discussed suitably, for example, in terms of the adatom-induced Majorana vortex as described in Fan et al., Nat. Comm 12, 1348 (2021). By the way, there are no STNM results on a YSR impurity in the supplementary information although the results away from the YSR impurity are presented (Fig. S6e and S6f).

There might be a misunderstanding regarding the probability of coinciding YSR and vortex state. In general, YSR states can be distinguished at zero field from vortex bound states. It is important to note that YSR states not only appear exactly at a magnetic impurity, but also away from the impurity. As we can see in Fig. S6 or Ref. 18, the YSR states extend ~ 8 nm from the impurity in $\text{FeTe}_{0.55}\text{Se}_{0.45}$, which is comparable to the size of a vortex core (coherence length ~ 3 nm). Moreover, YSR states disperse spatially in a similar way as CdGM states. In the presence of magnetic field, as the density of impurities in $\text{FeTe}_{0.55}\text{Se}_{0.45}$ is naturally high, the coincidence of YSR states and a vortex is actually not of small probability, because superconducting vortices tend to be bound at these impurities. Therefore, it is necessary to compare YSR states with vortex bound states, and in our case, to exclude the possibility that YSR states hide beneath a vortex core.

We have added a sentence in the main text on Page 7 to address this reason for comparing YSR and CdGM states.

We did measure a noise spectroscopic image (Fig. S6c) across the YSR impurity, which shows no contrast between on and off the impurity. Direct noise spectroscopy on YSR impurity is more complicated as discussed in Ref. 50, where not only strong coupling between the tip and impurity causes mechanical instability, but also other processes such as multipath tunneling and current-driven spin flip can happen.

(2) In page 6, from line 169 to 189, they describe the results obtained far away from a vortex (corresponding results are Fig. 4a and 4b, colored in blue). Although they mentioned, "at the gap energy, a step is visible", it is very difficult for me to find any signature of a step in the figures either Fig. 4a, b or Fig. S5a, b at the gap energy. But, there are broad increases at around $\pm(\Delta_T + \Delta_S)/2$.

We thank the reviewer for pointing this out. The broadened step is clearer in the extracted effective charge spectra in Fig. S5. We have changed the sentence to be more accurate.

(3) Although the typical tunneling spectra obtained at a YSR impurity and at off-impurity are clearly distinguishable (Fig. S6d), what would be benefits to utilize the STNM technique, which does not seem to be sensitive to the localized bound states as demonstrated in Fig. S6b and S6c?

We first point out that the effective charge tunneling into YSR states is theoretically $2e$ (Refs. 17 and 50), which is the same as tunneling into the bare superconductor. Therefore, it is not the insensitivity, but rather the sensitivity of our STNM technique that gives the expected result of $\sim 2e$ tunneling into YSR states.

(4) In page 6, line 179: the authors claim q^* reaches $1.97e$ at $\pm\Delta_T$. However, there are significant fractions locating rather close to the line $q^*=1e$ inside a superconducting gap ($\Delta_T \sim \Delta_T + \Delta_S$) for the results away from a YSR impurity (Fig. S6e and f), while the results on Pb(111) from the same group are more convincing for this claim (Ref. 36). What could be the reason for the significant fraction of $q^*=1e$ inside superconducting gap in this material without a magnetic field?

We note that the coherence peaks of Pb(111) are well-defined at $\pm\Delta_S = 1.35\text{meV}$, therefore the transition from $q^*=1e$ to $q^*=2e$ is sharp at $\pm(\Delta_T + \Delta_S)$. However, for $\text{FeTe}_{0.55}\text{Se}_{0.45}$ the coherence peaks are not well-defined [PRB 85, 094506]. We suspect that this could be the reason why the transition from $q^*=1e$ to $q^*=2e$ is broad in the range of $\pm(\Delta_T \sim \Delta_T + \Delta_S)$.

(5) Concerning descriptions for any transitions from $q^*=1$ to $q^*>1$ in this manuscript, it seems that such transitions do not occur at the expected energy positions. On the other hand, the authors tend to stick the q^* values right at the $\pm\Delta_T$ to deduce what they want to emphasize. As the superconducting tip has been used, the substantial contribution of the density of states at the tip can be involved in q^* values at $\pm\Delta_T$ as well. How do they discriminate the contribution of superconducting tip?

The transitions are more clearly visible in the effective charge spectra in Fig. S5, where outside the vortex q^* increases from $1e$ within $\pm(\Delta_T + \Delta_S)$, and inside the vortex this increase occurs in the range smaller than $(-\Delta_T, +\Delta_T)$.

The density of states of the tip will not influence the noise, because noise is measured per electron tunneling. We assume the question is about the energy at which it is measured.

We chose to measure at $\pm\Delta_T$ because we are interested in the state at the Fermi level of the sample. At this bias energy, as shown in Fig. S2a, the current comes from the resonant tunneling between the quasiparticle in one coherence peak of the tip's density of states and the zero-energy state of the sample (See Supplementary Note 2).

(6) The author's hypothesis in line 181 makes the STNM approach doubtful because it is unavoidable to investigate any vortex states without applying a magnetic field, meaning that a charge transfer of $q^*=1e$ contribution always exists inside the superconducting gap. Since the authors provide a significant contribution of the $q^*=1e$ process when two processes coexist as described in supplementary note 5, this

is quite serious. It seems this makes even more difficult in the unambiguous distinction between non-trivial states and trivial states.

Our hypothesis does suggest that a small fraction ($\sim 1\%$) of $q^*=1e$ process significantly lowers the total effective charge. However, as shown in Fig. 4b&d, we have been able to distinguish between $q^*=1e$ and $q^*=1.3e\sim 1.6e$, demonstrating the strength of STNM.

Regarding the unambiguous distinction between non-trivial states and trivial states: our results show for trivial YSR states $q^*=2e$, while for putative Majorana states and trivial CdGM states $q^*=1e$. We certainly have demonstrated by STNM we can distinguish between trivial YSR and putative Majorana states. For Majorana and CdGM states, because in principle we do not expect any difference, and our results do show the same effective charge. Therefore, the difficulty of our STNM measurements depends on the expected difference in q^* between nontrivial and trivial states.

(7) In this work, the Fano factor is redefined to emphasize the q^* -behavior. On the other hand, in the recent work utilizing the STNM for the YSR states in NbSe2 (Ref. 46), the coherency of the tunneling processes into the superconductors has been described based on the conventional definition of the Fano factor. I just wonder if the interpretations of the discrepancies from the expected results in this study can be improved or not in terms of the coherency in the tunneling process.

We would like to point out that, as stated in the Methods section, there is not a single, but multiple conventional definitions of the Fano factor that can be found in the literature. We have not redefined the Fano factor, but we included the charge of carriers (q) and their correlation (in any definition of Fano factor F) in effective charge q^* . The 'effective Fano factor' $F^*=qF/e$ described throughout Ref. 46 (now Ref. 50) is actually a redefined Fano factor, which is identical to q^*/e in our case.

(8) The authors mention that the STM junction is well in the single-channel. How do they confirm this since the valence orbitals contributing to the number of channels in the tunneling into both NbSe2 and FeTeSe might be different?

We meant our STM junction has a low transmission $\tau \ll 1$, thus the $(1 - \tau)$ term in the Landauer-Büttiker formalism [$(1 - T_n)$ in Eq. 57 of Ref. 37], can be well approximated by unity. We further note that if one would assume that the electrons could tunnel into different orbitals and count them as different channels, the total transmission $\tau \ll 1$ is controlled in our experiment and the individual transmission T_n is even smaller than τ , and thus our shot noise results do not change. Therefore, our STM junction can be well understood in the single-channel, low-transmission regime.

Minor comments:

(9) Line 152, Ref. 44 was cited as an example for the existence of mimicking zero-energy states in the non-conventional CdGM state. However, Ref. 44 is the study on the conventional CdGM states in a conventional superconductor (La).

We realized after Reviewer #2's point 7 that even conventional CdGM, such as observed in the original Ref. 44 by Kim et al. (Now Ref. 12), can appear at zero-energy when realistic conditions such as chemical disorder, magnetic field inside the vortex core, or Fermi surface of the superconductor are taken into

consideration. Following the reviewer's suggestion, in the revised manuscript, we now refer to them just as CdGM states, instead of 'non-conventional' CdGM states.

(10) Line 262, As STM experimentalists, do the authors seriously consider the proposal of a two-tip shot noise measurement setup where each tip tunnels into one localized vortex state? Please note that the application of the multi probe STM technique even for the nanostructures on the surface is almost impossible at present if the targeted areas are separated by a distance less than the physical radius of the individual probe. The authors should provide the outlook which can be challenging. Combining the shot-noise experiment with the spin-polarized STM technique (Wang et al., PRL 126, 076802 (2021) and A. Burtzloff et al., PRL 114, 016602 (2015)) would be a more reliable approach, which may be possibly complemented with each other.

We first note that the two-tip STM setup does not need to probe two neighboring Majorana-carrying vortices, but any two should show the nonlocal Majorana property, which can have a separation greater than the radius of the individual probe. Therefore, one just needs to use each tip to find one Majorana bound state individually, and then perform the noise measurements to test their cross-correlation. Second, it has already been demonstrated that multi-tip STM works with a probe spacing below 100 nm [e.g. APL 95, 052110 and Nanotechnology 30, 335702]. The nonlocal tunneling characteristics will be a stronger signature for Majorana bound states than the local tunneling characteristics, such as spin-polarized noise correlation (Refs. 53&54). We now also mention spin-polarized STM in the last paragraph of the revised manuscript.

REVIEWER COMMENTS

Reviewer #1 (Remarks to the Author):

Private communication sent only to Editor.

Reviewer #2 (Remarks to the Author):

I have read the response of the authors and I am satisfied with their answers. For this reason, I recommend publication.

Reviewer #3 (Remarks to the Author):

Dear Authors,

I thank the authors for the very detailed response to my comments and for having taken into consideration some of my arguments, as well as answered my questions and correct my misunderstanding. I am almost satisfied with their comments and the revised manuscript except for some points. I would like to recommend this manuscript for publication in Nature Communication after addressing a few remained issues described below:

Comment for (1)

"It is important to note that YSR states not only appear exactly at a magnetic impurity, but also away from the impurity."

"We did measure a noise spectroscopic image (Fig. S6c) across the YSR impurity, which shows no contrast between on and off the impurity. Direct noise spectroscopy on YSR impurity is more complicated as discussed in Ref. 50, where not only strong coupling between the tip and impurity causes mechanical instability, but also other processes such as multipath tunneling and current-driven spin flip can happen."

I understand what they want to claim for the reason why they try to exclude YSR states as the origin of the zero-bias conductance peak. But, it should be also noted that the intensity of the YSR state in conventional STS measurements has a maximum at the impurity and it is extended away from the impurity with decaying the intensity. If one can compare the LDOS maps with and without applying magnetic field, they can be clearly distinguishable unless they appear at the exact spatial locations. For example, it was not necessary for authors to discuss possible vortex bound states for the observed in-gap states in Ref. 18. Therefore, I am still struggling to understand the benefit of using STNM technique even it is quite difficult to measure it at the impurity as they commented.

They cited Ref. 50 to discuss the 2e transfer for the YSR states (Line 220~221). In Ref. 50, the authors claim that the concomitant contribution of the 1e and 2e transfers simultaneously through the YSR states. Moreover, noise spectroscopic results for YSR impurities in NbSe₂ ($F^* < 1$ or $F^* > 1$) and bare substrates ($F^* = 1$) show a clear difference, starkly contrasting to the present work. I suggest considering an extended discussion about the discrepancy to rationalize their interpretation. For example, why is the 1e-transfer absent for the YSR state in FeTe_{0.55}Se_{0.45}, and so on?

Comment for (5)

"The density of states of the tip will not influence the noise, because noise is measured per electron tunneling. We assume the question is about the energy at which it is measured."

They argue that the broadening of transition from $q^* = 1e$ to $q^* = 2e$ might be due to the "not well-defined coherence peak" for FeTeSe in the comment for (4). This could be the same for the tip that is one electrode as well in the junction. Because the authors prepared the superconducting Pb tip by indenting the tip into Pb(111) surface, which, in general, generates various superconducting tips depending on the circumstance of the tip-apex (PRB 74 132501 (2006)), which is not visible in SN junction (such as Au(111)). I wonder if the difference from the results in Ref. 50 can be due to the superconducting tip or not.

By the way, in the deconvolution procedure, how did the authors deconvolute the conductance spectra obtained under the magnetic field since there is no suitable formulation considering the magnetic field?

Comment for (10)

"We first note that the two-tip STM setup does not need to probe two neighboring Majorana-carrying vortices, but any two should show the nonlocal Majorana property, which can have a separation greater than the radius of the individual probe. Therefore, one just needs to use each tip to find one Majorana bound state individually, and then perform the noise measurements to test their cross-correlation."

My concern was basically because of the sentence in the manuscript, "A further proposal in the low-bias limit suggested a two-tip shot noise measurement setup where each tip tunnels into one localized vortex state,... (Line 267~268). I misunderstood that two tips should access a single vortex. I thank the authors for the detailed interpretations and maybe the sentence can be specified a little further to avoid misunderstanding.

Concerning the previous efforts to distinguish the Majorana vortex states from the trivial ones, the references as follows might be relevant.

Takuto Kawakami et al., Phys Rev Lett 115, 177001 (2015).

Hao-Hua Sun et al., Phys Rev Lett 116, 257003 (2016).

Point-by-point reply to the Reviewers' comments

We thank all reviewers for endorsing publication in Nature Communications. Here we reply to Reviewer #3's new comments, and we marked all new changes in **green** for this revision in the manuscript. Below is a summary of the changes:

- **Main text: we changed the last sentence on Page 8 as suggested by Reviewer #3.**
- **Main text: we added the two references (Refs. 53 and 54) suggested by Reviewer #3.**
- **Supplemental Materials: we added a paragraph to Note 4 to compare our results with Ref. 50, as suggested by Reviewer #3.**

Additionally, we performed small changes marked in **green** in the main text, such as rewording sentences and figure captions.

With kind regards,

Milan Allan, on behalf of the authors

Point-by-point reply to Reviewers.

Reviewer #3

Dear Authors,

I thank the authors for the very detailed response to my comments and for having taken into consideration some of my arguments, as well as answered my questions and correct my misunderstanding. I am almost satisfied with their comments and the revised manuscript except for some points. I would like to recommend this manuscript for publication in Nature Communication after addressing a few remained issues described below:

We thank the reviewer for the comments. We have addressed all the remaining points below and made changes to the manuscript. We thank the reviewer for recommending the publication of our work in Nature Communications after addressing the remaining issues.

Comment for (1)

"It is important to note that YSR states not only appear exactly at a magnetic impurity, but also away from the impurity."

"We did measure a noise spectroscopic image (Fig. S6c) across the YSR impurity, which shows no contrast between on and off the impurity. Direct noise spectroscopy on YSR impurity is more complicated as discussed in Ref. 50, where not only strong coupling between the tip and impurity causes mechanical instability, but also other processes such as multipath tunneling and current-driven spin flip can happen."

I understand what they want to claim for the reason why they try to exclude YSR states as the origin of the zero-bias conductance peak. But, it should be also noted that the intensity of the YSR state in conventional STS measurements has a maximum at the impurity and it is extended away from the impurity with decaying the intensity. If one can compare the LDOS maps with and without applying magnetic field, they can be clearly distinguishable unless they appear at the exact spatial locations. For example, it was not necessary for authors to discuss possible vortex bound states for the observed in-gap states in Ref. 18. Therefore, I am still struggling to understand the benefit of using STNM technique even it is quite difficult to measure it at the impurity as they commented.

As mentioned, because the density of impurities in $\text{FeTe}_{0.55}\text{Se}_{0.45}$ is naturally high, the coincidence of YSR states and a vortex is actually not of small probability, especially because superconducting vortices tend to be bound at these impurities. Therefore, we believe it is worthwhile to compare YSR states with vortex bound states by another method to exclude the possibility that YSR states hide beneath a vortex core. Our local shot noise measurement does work for this purpose.

They cited Ref. 50 to discuss the 2e transfer for the YSR states (Line 220~221). In Ref. 50, the authors claim that the concomitant contribution of the 1e and 2e transfers simultaneously through the YSR states. Moreover, noise spectroscopic results for YSR impurities in NbSe_2 ($F^* < 1$ or $F^* > 1$) and bare substrates ($F^* = 1$) show a clear difference, starkly contrasting to the present work. I suggest considering an extended

discussion about the discrepancy to rationalize their interpretation. For example, why is the $1e$ -transfer absent for the YSR state in $\text{FeTe}_{0.55}\text{Se}_{0.45}$, and so on?

We would like to clarify that Ref. 50 explicitly writes “... we obtain $F^* = 1$ for all voltages where the current is large enough to detect shot noise accurately [Fig. 1(d)] as expected for quasiparticle tunneling following Poissonian statistics”. For tunneling into the bare superconductor, Ref. 50 only measures F^* outside the gap, where $F^*=1$ is expected. However, at the energy of the YSR states on NbSe_2 , i.e. ~ 0.5 meV, they do not have data for F^* of the bare NbSe_2 because their current is too low and the corresponding shot noise is below their resolution. In principle, considering a normal tip used in this case, inside the gap $F^*>1$ and increases gradually to 2, see e.g. Ref. 42 for a different superconductor (sample). Therefore, there is no contradiction between Ref. 50 and our data for YSR states, and our use of a superconducting tip helps to measure the effective charge more accurately.

Ref. 50 has explained that Andreev reflection requires particle-hole symmetric resonances. The smaller of the asymmetric resonances leads to a deficiency of providing particle (or hole) for the Andreev ($2e$) process, so that tunneling into the excess hole (or particle) component in the larger resonance has to be mediated by an inelastic quasiparticle relaxation process. In our experiment this inelastic quasiparticle relaxation ($1e$) process is strongly suppressed because: i) our YSR states in $\text{FeTe}_{0.55}\text{Se}_{0.45}$ are almost symmetric in amplitude, compared to the asymmetric peaks of YSR states in NbSe_2 in Ref. 50; ii) we are in the strong tunneling regime (our current 800 pA is much larger than the threshold ~ 90 pA, see Supplementary Note 4 for the calculation), compared to Ref. 50 in the weak tunneling regime (20 pA tunneling current with a similar threshold) where single-electron tunneling dominates.

We have included the above discussion in Supplementary Note 4, as it only involves details about YSR states instead of Majorana states.

Comment for (5)

"The density of states of the tip will not influence the noise, because noise is measured per electron tunneling. We assume the question is about the energy at which it is measured."

They argue that the broadening of transition from $q^*=1e$ to $q^*=2e$ might be due to the "not well-defined coherence peak" for FeTeSe in the comment for (4). This could be the same for the tip that is one electrode as well in the junction. Because the authors prepared the superconducting Pb tip by indenting the tip into $\text{Pb}(111)$ surface, which, in general, generates various superconducting tips depending on the circumstance of the tip-apex (PRB 74 132501 (2006)), which is not visible in SN junction (such as $\text{Au}(111)$). I wonder if the difference from the results in Ref. 50 can be due to the superconducting tip or not.

We would like to clarify that, the gap of the superconducting (Pb or Nb) tip is always defined in an individual experiment, even in the reference [PRB 74 132501 (2006)] provided; the Δ_1 and Δ_2 mentioned in this reference are *not* two tip gaps but the gaps of the sample (bulk Nb) and the Nb tip, respectively. We have, in addition to $\text{Au}(111)$, characterized our tip on a Pb sample (Fig. S1a), and made sure that the quasiparticle coherence peak of our tip has a width ~ 0.25 meV, which is much smaller than the step width ~ 1 meV in q^* . The latter is more consistent with the energy range of the various gap sizes reported for

FeTe_{0.55}Se_{0.45} [1.4-2.4 meV, see Science 367, 104(2020)]. However, we do not want to claim the origin of the broadness as we lack solid experimental evidence to distinguish the origins.

By the way, in the deconvolution procedure, how did the authors deconvolute the conductance spectra obtained under the magnetic field since there is no suitable formulation considering the magnetic field?

We explain our deconvolution procedure clearly in Supplementary Note 1. We first measure dI/dV on Au(111) at different fields as shown in Fig. S1b, then following Eq. S1, which is a general formula for tunneling conductance (regardless of the existence of field), we extract the density of states of the tip assuming a phenomenological (Dynes) model and a constant density of states (DOS) for the Au(111) in the energy range from -10 meV to +10 meV. The fitting parameters for Δ_t and Γ are shown in Fig. 1c. For a certain field, we use the corresponding values of Δ_t and Γ to get tip DOS (N_t) by the Dynes model in Eq. S1, to extract the sample DOS for FeTe_{0.55}Se_{0.45}.

Comment for (10)

"We first note that the two-tip STM setup does not need to probe two neighboring Majorana-carrying vortices, but any two should show the nonlocal Majorana property, which can have a separation greater than the radius of the individual probe. Therefore, one just needs to use each tip to find one Majorana bound state individually, and then perform the noise measurements to test their cross-correlation."

My concern was basically because of the sentence in the manuscript, "A further proposal in the low-bias limit suggested a two-tip shot noise measurement setup where each tip tunnels into one localized vortex state,... (Line 267~268). I misunderstood that two tips should access a single vortex. I thank the authors for the detailed interpretations and maybe the sentence can be specified a little further to avoid misunderstanding.

We thank the reviewer for pointing out the confusing sentence, and we have revised it to make it clearer.

Concerning the previous efforts to distinguish the Majorana vortex states from the trivial ones, the references as follows might be relevant.

Takuto Kawakami et al., Phys Rev Lett 115, 177001 (2015).

Hao-Hua Sun et al., Phys Rev Lett 116, 257003 (2016).

We thank the reviewer for providing the references and we have included them as Refs. 53-54 in the revised manuscript.

REVIEWERS' COMMENTS

Reviewer #3 (Remarks to the Author):

I thank the authors for the detailed explanations to my comments and for having taken into consideration some of my arguments. The revised version is in a good shape and recommend for acceptance.